

**Using flow cytometry and light-induced fluorescence technique to characterize the variability and characteristics of bioaerosols in springtime at Metro Atlanta, Georgia**

Arnaldo Negron[a,b], Natasha DeLeon-Rodriguez[c$], Samantha M. Waters[a#], Luke D. Ziemba[d], Bruce Anderson[d], Michael Bergin[e], Konstantinos T. Konstantinidis[f,c*], and Athanasios Nenes[a,b,g,h,i*]

[a] School of Earth and Atmospheric Sciences, Georgia Institute of Technology, Atlanta, GA 30332, USA
[b] School of Chemical and Biomolecular Engineering, Georgia Institute of Technology, Atlanta, GA 30332, USA
[c] School of Biology, Georgia Institute of Technology, Atlanta, GA 30332, USA
[d] Chemistry and Dynamics Branch/Science Directorate, National Aeronautics and Space Administration Langley Research Center, Hampton, VA 23681, USA
[e] Department of Civil and Environmental Engineering, Duke University, Durham, NC, 2770, USA
[f] School of Civil and Environmental Engineering, Georgia Institute of Technology, Atlanta, GA 30332, USA
[g] Institute of Environmental Research & Sustainable Development, National Observatory of Athens, GR-15236, Greece
[h] Institute for Chemical Engineering Science, Foundation for Research and Technology Hellas, Patra, GR-26504, Greece
[i] Laboratory of Atmospheric Processes and their Impacts (LAPI), School of Architecture, Civil & Environmental Engineering, Ecole Polytechnique Fédérale de Lausanne, CH-1015, Switzerland
[$] Currently at: Puerto Rico Science, Technology and Research Trust, Rio Piedras, 00927, Puerto Rico
[#] Currently at: Department of Marine Sciences, University of Georgia, Athens, GA 30602-3636
*Corresponding Author

## Abstract

The abundance and speciation of primary biological aerosol particles (PBAP) is important for understanding their impacts on human health, cloud formation and ecosystems. Towards this, we have developed a protocol for quantifying PBAP collected from large volumes of air with a portable wet-walled cyclone bioaerosol sampler. A flow cytometry (FCM) protocol was then developed to quantify and characterize the PBAP populations from the sampler, which were confirmed against epifluorescence microscopy. The sampling system and FCM analysis were used to study PBAP in Atlanta, GA over a two-month period and showed clearly defined populations of DNA-containing particles: Low Nucleic Acid-content particles (bioLNA), High Nucleic Acid-content particles (HNA) being fungal spores and pollen. We find that daily-average springtime PBAP concentration (1 to 5µm diameter) ranged between $1.4\times10^4$ and $1.1\times10^5$ m$^{-3}$. The BioLNA population dominated PBAP during dry days (72 ± 18%); HNA dominated



the PBAP during humid days and following rain events, where HNA (e.g., wet-ejected fungal spores)
comprised up to 92% of the PBAP number. Concurrent measurements with a Wideband Integrated
Bioaerosol Sensor (WIBS-4A) showed that FBAP and total FCM counts are similar; HNA (from FCM)
significantly correlated with ABC type FBAP concentrations throughout the sampling period (and for the
same particle size range, 1-5 µm diameter). However, the FCM bioLNA population, possibly containing
bacterial cells, did not correlate to any FBAP type. The lack of correlation of any WIBS FBAP type with
the bioLNA suggest bacterial cells may be more difficult to detect with autofluorescence than previously
thought. Identification of bacterial cells even in the FCM (bioLNA population) is challenging, given that
the fluorescence level of stained cells at times may be comparable to that seen from abiotic particles. HNA
and ABC displayed highest concentration on a humid and warm day after a rain event (4/14), suggesting
that both populations correspond to wet-ejected fungal spores. Overall, information from both instruments
combined reveals a highly dynamic airborne bioaerosol community over Atlanta, with a considerable
presence of fungal spores during humid days, and a bioLNA population dominating bioaerosol community
during dry days.

**Introduction**

Primary biological atmospheric particles (PBAP), also called bioaerosols, are comprised of airborne
microbial cells (e.g. bacteria, diatoms), reproductive entities (e.g. pollen, fungal spores), viruses and
biological fragments. Bioaerosols are ubiquitous, with potentially important impacts on human health,
cloud formation, precipitation, and biogeochemical cycles. (Pöschl, 2005; Hoose et al., 2010; DeLeon-
Rodriguez et al., 2013; Morris et al., 2014; Longo et al., 2014; Fröhlich-Nowoisky et al., 2016;
Myriokefalitakis et al., 2016). Despite their low number concentration relative to abiotic particles, PBAP
possess unique functional and compositional characteristics that differentiate them from abiotic aerosol.
For example, certain PBAP constitute the most efficient of atmospheric ice nucleators, affecting the
microphysics of mixed phase clouds and precipitation (Hoose and Möhler, 2012; Sullivan et al., 2018). The
mass and nutrient content of PBAP may suffice to comprise an important supply of bioavailable P to
oligotrophic marine ecosystems (Longo et al., 2014; Myriokefalitakis et al., 2016). In addition, the
concurrence of disease outbreaks during dust storms has been attributed to pathogenic microbes attached
to airborne dust that are subsequently inhaled (Griffin et al., 2003; Ortiz-Martinez et al., 2015; Goudie

2014).

Quantification of the concentration and size of PBAP is critical for understanding their environmental
impacts. Measuring PBAP however poses a challenge for established microbiology tools, owing to their
low atmospheric concentration ($10^3$ - $10^6$ cells m$^{-3}$ air; Fröhlich-Nowoisky et al., 2016) and wide diversity
of airborne particle types and sizes. For instance, only a fraction of microorganisms (an estimated 5%; Chi



and Li et al., 2007) can be cultured, and cultivation cannot be used to quantify dead organisms, viruses or
fragments, while most culture-independent methods are optimized for more abundant microbial
populations. Epifluorescence microscopy (EPM) is the standard for bioaerosol quantification but is not
high-throughput and requires considerable time for quantification of concentration per sample. Flow
cytometry (FCM) is an analysis technique based on the concurrent measurement of light scattering and
fluorescence intensity from single particles (Wang et al., 2010). FCM requires a liquid suspension of
bioparticles that flows through an optical cell and interrogated with a series of laser beams. Each sample is
pretreated with stains that targeting specific macromolecules (e.g. DNA/RNA) which subsequently
fluoresce when excited by the FCM lasers. The resulting scattering and fluorescent light emissions are then
detected by an array of sensors to allow the differentiation of biological and abiotic (e.g. dust) particles
according to the characteristic specific to the stain used. FCM has proved to be as reliable as EPM, but with
the advantage of lower uncertainty, higher quantification efficiency and requiring considerably less time
and effort than EPM per sample (Lange et al., 1997). FCM is frequently used in biomedical research to
quantify eukaryotic cell populations, and in microbiology to quantify a wide variety of yeast and bacterial
cells (Nir et al., 1990; Van Dilla et al., 1983). FCM is also used to study environmental samples, e.g., to
differentiate low nucleic acid (LNA) from high nucleic acid (HNA) phytoplankton in aquatic environments
(Wang Y. et al 2010; Müller et al., 2010). Despite its advantages, FCM has seen little use in the bioaerosol
field to date (e.g., Chen and Li, 2005; Liang et al., 2013), owing in part to the challenges associated with
collecting sufficient PBAP mass for sufficient counting statistics to be obtained (Chen and Li, 2005; Liang
et al., 2013). Chen and Li (2005) determined that for counting purposes, the SYTO-13 nucleic acid stain is
the most effective (among five different nucleic acid stains studied) for determining reliable concentration
of bioaerosols.
The SYTO-13 stain can also be used to provide insights on the stress/metabolic state of microbes.
Guindulian et al. (1997) used FCM to study biological particles in fresh and starved seawater samples
collected from the West Mediterranean Sea (Spain). Samples were analyzed either immediately or after
starvation for 2-3 days. Also, *E.coli* pure cultures were tested before and after starvation in sea water. In
both situations, samples were treated with DNAse and RNAse and subsequently stained with SYTO-13 to
measure cellular DNA content (starvation ensures that intracellular RNA is negligible in marine
populations, so that the SYTO-13 intensity is directly related to the DNA content of cells). Guindulian et
al. (1997), with starved seawater samples and *E.coli* pure cultures together suggest that the stress level
caused by marine starvation reduces RNA content in aquatic microorganisms to an undetectable level. This
has important implications for the detection of atmospheric PBAP, as cells are exposed to multiple stressors
when airborne.



Light Induced Fluorescence (LIF) is an increasingly utilized technique for bioaerosol quantification, and it relies on measuring the autofluorescence intensity of specific high yield fluorophores (e.g., Nicotinamide Adenine Dinucleotide – NADH co-enzyme, flavins and amino acids like Tryptophan and Tyrosine) present in PBAP. The major advantage of the technique is that it is fully automated, does not require a liquid suspension (i.e., it directly senses particles suspended in air) and that provides high frequency measurements (~1 Hz) which make it ideal for monitoring and bioaerosol quantification. Particles detected by LIF, called Fluorescent Biological Atmospheric Particles (FBAP), although not equal to PBAP, may still constitute a large fraction of the biological particles (Healy et al., 2014; Gosselin et al., 2016). Using LIF, FBAP diurnal cycles showing maximum concentrations during evenings and minimum around middays, especially in heavily vegetated environments have been observed. This behavior has been related to known temperature and relative humidity release mechanism of certain fungal spore species (Wu et al., 2007; Gabey et al., 2010; Tropak and Schnaiter, 2013). Huffman et al. (2010) used a UV-Aerodynamic Particle Sizer (UV-APS) to show that the concentration and frequency of occurrence of 3µm FBAP particles at Mainz, Germany (semi-urban environment) exhibited a strong diurnal cycle from August through November: with a first peak at ~ $1.6 \times 10^4$ m$^{-3}$ at mid-morning (6-8 am) followed by a constant profile (~ $2\text{-}4 \times 10^4$ m$^3$) throughout the rest of the day. Similar studies in urban and densely vegetated environments suggest a notable difference in the size distributions, diurnal behavior and FBAP loading between the two environments. Gabey et al., 2011 found that the FBAP in Manchester, UK follow a characteristic bimodal distribution with peaks at 1.2µm and 1.5 - 3.0 µm. As in Mainz, the concentration of larger particles peaks in the mid-morning, ranges from 0 to 300 L$^{-1}$, and the 1.2µm peak is linked to traffic activity. However, at the Borneo tropical rain forest FBAP concentrations peak during the evening with a robust 2-3µm population and concentrations ranging from 100 to 2000 L$^{-1}$ (Gabey et al., 2010).

LIF-based observations (e.g. UV-APS, WIBS), combined with measurements of molecular tracers (e.g. mannitol and arabitol) and endotoxin measurements provide a more complete picture of PBAP emissions. Gosselin et al. (2016) applied this approach during the BEACHON-RoMBAS field campaign. A clear correlation between FBAP and the molecular markers is seen, indicating an increase of fungal spores during rain events. FBAP concentrations and molecular marker-inferred (arabitol and mannitol) fungal spore concentrations (1.7pg mannitol per spore and 1.2 pg arabitol per spore; Bauer et al., 2008) were within the same order of magnitude. The UV-APS FBAP concentration during rain events was higher than the fungal spore concentrations inferred from the concentration of molecular markers, which suggest other non-fungal spore fluorescent particles are detected as well as fungal spores by the UV-APS. In the same study, the WIBS-3 cluster (determined using Crawford et al., 2015) linked to fungal spores gave concentrations that were 13% lower than those derived from molecular marker concentrations during rain events. During dry events, FBAP and molecular markers derived fungal spore concentrations were poorly correlated. It is



currently unknown the degree to which all types of PBAP are consistently detected by LIF over different
time of the year and different environments; it is likely, however, that for certain classes of bioparticles
(e.g., pollen and fungi) the detection efficiency using LIF is relatively high. However, the low intrinsic
fluorescence intensity of bacteria and high variability of thereof in relation to metabolic state may lead to
their misclassification as non-biological particles (Hernandez et al., 2016).

For LIF-based quantification of PBAP to be effective, it requires the intrinsic fluorescence of biological

material to exceed that of non-biological matter. Depending on the type, metabolic state and species, PBAP
autofluorescence may vary orders of magnitude and therefore LIF may not always be able to differentiate
between biological and abiotic particles. For example, Tropak and Schnaiter (2013) showed that laboratory-
generated mineral dust, soot and ammonium sulfate may be misclassified as FBAP. To address
misclassification, Excitation Emission Matrices (EEMs) have been developed for biomolecules (e.g.
tryptophan, tyrosine, riboflavin) and non-biological (e.g. Pyrene, Napthalane, Humic Acid) molecules.
EEMs provide the wavelength-dependent fluorescence emission spectra as a function of the excitation
wavelength and are used to assign spectral modes to known fluorophores. The structure of EEMs is
important for identifying molecules that are unique to PBAP and allow their identification by LIF; it is this
principle upon which detectors in commercial FBAP measurements (e.g. WIBS, UV-APS) are based upon.
Comparison of EEMs from biological and non-biological molecules show that even when biomolecules
have higher autofluorescence intensity than non-biologicals in the LIF detection range, interferences from
non-biological compounds (e.g. polycyclic aromatic hydrocarbons and soot) from combustion emissions
can influence LIF detection (Pöhlker et al., 2012). Considerable work remains on determining which
detector(s) or combination thereof provides an unambiguous identification of bioaerosols and related
subgroups (e.g. bacteria, fungal spores, pollen). Towards this, an aerobiology catalog of pure cultures has
been developed for the WIBS-4 (Hernandez et al., 2016), where, (i) pollen and fungal spore species
autofluoresce much more than bacteria, and, (ii) bioaerosol subgroups are more successfully discriminated
by specific detector(s). However, the same study showed that instrument-to-instrument variability in
fluorescence detection poses a considerable challenge, as applying common detection thresholds across
instruments leads to considerable differences in PBAP concentration and composition.

Another important issue for LIF-based PBAP is the impact of atmospheric oxidants, UV and other

stressors on the fluorescence intensity of PBAP. This is important, given the prevalence of PBAP
throughout the atmosphere, including the extreme conditions in the upper troposphere (DeLeon-Rodriguez
et al., 2013). Pan et al. (2014) tested the effect of relative humidity and ozone exposure in the
autofluorescence spectra of octapeptide aerosol particles using an UV-APS connected to a rotating drum.
Octapeptides, organic molecules containing eight amino acids and present in cells, were used as a proxy to



study the aging of tryptophan and results suggest bioaerosols exposure to typical ozone concentrations
(~150ppb) decrease tryptophan fluorescence intensity and affects PBAP detection. Laboratory experiments
cannot always reproduce the wide variety of environmental conditions and stressors that can affect the
metabolism state of microbes, and hence their autofluorescence. Joly et al. (2015) studied the survival rate
of multiple bacterial (e.g. *Pseudomona syringae Sphingomonas sp. And Arthrobacter sp.*) and yeast (e.g.
*Dioszegia hungarica*) strains isolated from cloud water upon exposure to oxidants (e.g. $H_2O_2$), solar light
(e.g. UV radiation), osmotic shocks (e.g. multiple NaCl concentrations) and freeze-thaw cycles. Among
these stressors, the freeze-thaw cycles affected most the survival rate (quantified as the quotient of the
colony forming unit (CFU) counts before and after exposure to each stressor dose) of bacterial cells.
*Arthrobacter sp.* showed the lowest survival rates ($< 20\%$) per cycle, and the highest survival rate of all
bacterial strains was observed at $10^8$ cell mL$^{-1}$ (highest concentration), suggesting that high cell
concentrations lead to cell aggregation and provided protection against freeze-thaw cycles. The survival
rate of the yeast *Dioszegia hungarica* was mostly affected by UV radiation showing the effect of each
stressor in the survival rate of cells may depend on the characteristics of each cell. Even though the survival
rate and the intrinsic fluorescence intensity of bioaerosols have not been correlated, multiple stressors can
be affecting bioaerosols LIF detection and these issues regarding the use of LIF need to be resolved to fully
understand their PBAP detection efficiency over the wide range of atmospheric conditions and PBAP
population composition (Toprak and Schnaiter, 2013; Hernandez et al., 2016).

The aims of the study were to (i) develop an effective and reliable FCM detection and quantification
protocol; (ii) apply the protocol to understand bioparticle populations and their variability in the metro
Atlanta area during different meteorological conditions, and, (iii) compare FCM and WIBS-4A results to
have a better understanding of PBAP day-to-day variability. To our knowledge, this study is the first to
develop a FCM protocol to identify and quantify well-defined speciated bioaerosols populations from
samples collected from a modified state-of-the-art biosampler. LIF sampling of bioaerosol side-by-side
with established and quantitative biology tools (FCM and EPM) was conducted to assess the LIF detection
capabilities toward different bioaerosol populations and under atmospherically-relevant conditions during
this study. Atlanta is selected for PBAP sampling, as it provides a highly populated urban environment
surrounded by vast vegetative areas; this and the broad range of temperature and humidity ensures a wide
range of PBAP population composition, state and concentrations. All the samples collected are compared
side-by-side to concurrent WIBS-4A data collected over the same time period.






## 2. Instrumentation and Methodology

### 2.1 Bioaerosol Sampler

Sampling was performed using the SpinCon II (InnovaPrep LLC, Inc.) portable wet-walled cyclone aerosol sampler. Aerosol is collected by inertial impaction with a recirculating liquid film in the cyclone; evaporative losses of thr filmare compensated so that the sample volume is maintained constant during a sample cycle. The particle collection efficiency for 1µm, 3µm, 3.5µm and 5.0µm particles is about 47.3±2.1%, 56.1±3.9%, 14.6 ± 0.6 and 13.8 ± 2.2%, respectively (Kesavan et al., 2015). However, the experiments conducted using 1µm PSL and 3µm PSL, 3.5µm oleic acid and 5.0µm oleic acid particles not necessarily quantify the collection efficiency of biological particles in this size range. Even with a lower efficiency than any impingement sampler, SpinCon has a better performance (product of the flow rate and the sampling efficiency) than any impingement sampler due to its high volumetric flow rate, which make it more suitable for bioaerosols detection (Kesavan et al., 2015). The efficiency, power consumption and performance of 29 biosamplers were analyzed by Kesavan et al. (2015) to determine which are best suited for indoor or outdoor sampling. The study concluded biosamplers effectiveness will be determined by their performance in the size range of interest, rather than just by looking its sampling efficiency. Furthermore, Santl-Temkiv et al. (2017) recently studied SpinCon retention efficiency towards sea water heterogenous and pure cultured *P.agglomerans* populations ($\sim 10^5$ cells mL$^{-1}$) after 1 hr sampling period by comparing FCM-derived concentrations (using SYBR green stain) before and after the sampling period. SpinCon retains 20.6±5.8% of the *P.agglomerans* concentration, whereas 55.3±2.1% of the sea water microbial concentration is retained after sampling for 1hr.

In our study, the biosampler was run at 478L min$^{-1}$ for 4hr sampling cycles. Phosphate-buffered saline (PBS) 1X pH 7.4 solution was used and the instrument compensated evaporation by supplying Milli-Q water to maintain the PBS concentration constant. Upon termination of each sampling cycle, the instrument was programmed to dispense the sample in a 15mL centrifuge tube. Then, 10µl of formalin (37 wt.% formaldehyde) per mL of solution was added to every sample for preservation and samples were stored at 4°C. Given the long sampling times and the low concentration of PBAP, the fluid supply system of the instrument was modified and a cleaning protocol (CP) has been developed, which is described below.

SpinCon II H$_2$O and PBS supply bags were replaced by two 2L autoclavable Nalgene bottles (Thermo Scientific Inc.) with antimicrobial tubing, connectors and a small HEPA filter connected to vent and prevent coarse and submicron particles contamination (Figure 1). Bottles were autoclaved and filled with Milli-Q water and PBS, beforehand sterilized with 0.2µm pore bottle top filters (Thermo Fisher Inc.) and transferred inside a biosafety cabinet. An aliquot of each fluid obtained after preparation was evaluated for sterility by EPM and FCM.





The cleaning protocol (CP) of the biosampling system consists of two phases. During phase one, all
acrylic windows and the outside of the collector/concentrator were cleaned with ethanol 70 wt. %. Then,
the instrument inlet, outlet, and the inside of the collector/concentrator was cleaned with ethanol 70 wt. %.
In the second phase, the SpinCon II inlet was connected to a HEPA filter to provide a particle-free source
of air to the sampling system; the instrument was then washed with ethanol 70 wt.%, 10 wt.% bleach
solution, PBS and Milli-Q $H_2O$, respectively. The wash consisted of a rinse, a 2 minutes sample and filling
the instrument collector/concentrator with the fluid in use (i.e., bleach solution, ethanol, PBS and Milli-Q
$H_2O$). The collector/concentrator was drained after 1 minute. The above were repeated for the remaining
fluids, taking 5 minutes per fluid. Overall, the CP requires 45 minutes; upon completion, a blank is obtained
to constrain the residual contamination levels after cleaning (described below). Finally, the HEPA filter
was disconnected, instrument inlets and outlets were sealed and the inlet tube was cleaned with ethanol 70
wt.% to be ready for rooftop sampling. SpinCon II was rinsed with ethanol 70wt.% after each sampling
episode and the cleaning protocol was applied before each sample.
Several blanks were obtained to quantify the levels of PBAP contamination in the fluids and sampler,
and to ensure that they were sufficiently low to not bias the detection, identification and quantification of
the PBAP. Furthermore, an instrument blank was obtained after a CP to constrain residual particles, by
running the sampler for 2 minutes, while sampling air with a HEPA filter connected to the inlet of the
SpinCon II. Another blank was collected to characterize any contamination of biological particles from the
supply of PBS and water in the SpinCon II. This was done by operating the SpinCon II for a 4hr period
with a HEPA filter connected to the inlet which completely cleans the air entering the wet cyclone from
any bioparticles.  All blanks were analyzed directly via FCM (Sect. 2.3) and EPM.
The volumetric flow rate within the SpinCon II was routinely calibrated by a VT100 Hotwire Thermo-
anemometer (Cole Palmer Inc.) using a 3-hole round duct transverse approach. A 1 ¼" OD tube with the
same diameter as the SpinCon II inlet was designed with 3 holes. Each hole was 60° apart from the other
and the holes were perpendicular to the axial air flow direction of the tube. (Supplementary Information,
Figure S1). Triplicates of flow rate measurements were taken in each hole at the center of the tube and
averaged to determine SpinCon II volumetric flow rate ($478.0 \pm 6.4$ L min$^{-1}$).

## 261    2.2 Flow Cytometry

During this study, a BD Accuri C6 flow cytometer (BD Bioscience Inc.) was used for Flow Cytometry.
The instrument quantifies suspended cells in aqueous medium at three flow velocity modes (slow, medium
and fast flow at 14, 35 and 66 µL min$^{-1}$, respectively). It excites particles with a 488nm laser and possesses
four fluorescence detectors: FL1 ($533\pm30$nm), FL2 ($585\pm40$nm), FL3 ($> 670$nm) and FL4 ($675\pm25$nm),
which make it possible to analyze the fluorescence from multiple dyes concurrently. In this study, 2.5 µM



SYTO-13 nucleic acid probe was added to the fixed samples and incubated for 15min in the dark at room
temperature to stain biological particles. Additionally, 10µL of 15µm polystyrene bead suspension was
added to the 1mL total volume samples as an internal standard for PBAP concentration and size
quantification. The BD Accuri C6 was cleansed before each use with 0.2µm filtered Milli-Q water in fast
mode for 10min; background particle counts were typically reduced to $1\mu L^{-1}$. At the beginning of every
experiment, a 1mL blank of the atmospheric sample without SYTO-13 and beads was analyzed, used in
quantification calculations (Sect. 3.1). Each sample was run in slow mode for 5min. After each sample, the
instrument was flushed with 0.2µm filtered Milli-Q water in slow flow for 1 minute (important for robust
quantification of the typically low concentrations of the atmospheric samples). SYTO-13 fluorescence
intensity was quantified by the FL1-A detector and used in combination with other parameters (FSC-A &
SSC-A) to constrain the PBAP populations present. FSC-A measured forward ($0° \pm 13°$) scattering and is
used to characterize the size of particles; SSC-A measured the side ($90° \pm 13°$) scattering and is used to
characterize the internal complexity (non-sphericity/shape) of particles. A 80,000 unit intensity FSC-H
threshold (default FSC-H threshold value suggested by the manufacturer to minimize the effect of noise)
was set in the instrument during data acquisition to minimize the effects of noise on bioparticle counts. The
FSC-H channel (where H denotes height), measures single-particle forward scattering (FSC) intensity based
on the peak (maximum point) of the voltage pulse curve recorded when a single particle goes through the
interrogation point in the flow cytometer, whereas FSC-A, where A denotes area, measures single-particle
FSC intensity based on the area below the curve of the recorder pulse. When the 80,000 unit FSC-H
threshold is defined, only signals with an intensity greater than or equal to threshold value will be processed,
and this could affect the statistics and detection efficiency of the flow cytometer toward small particles ($\leq$
1µm). Experiments conducted with 1.0µm polystyrene beads suspension (Supplemental information;
Figure S16) have shown that 1.0µm beads have FSC-H intensities above the 80k threshold, no particle
losses is observed, and beads estimated concentration agree with the reported by the manufacturer (~6 ×
$10^7$ $mL^{-1}$; Life Technologies, Inc.) The FCM data from each sample was analyzed using the Flow Jo
software (https://www.flowjo.com/solutions/flowjo) to gate and quantify bioparticles population. The same
procedure was used to analyze the PBS, Milli-Q water and blanks.
**2.3 LIF detection of PBAP**

The WIBS-4A (referred to henceforth as "WIBS") is a single biological particle real time sensor, which

measures particle light scattering and autofluorescence in an approximately 0.5 – 15µm particle range
(www.dropletmeasurement.com). Particles are initially sized using the 90-degree side-scattering signal
from a 635 nm continuous-wave diode laser. The scattering intensity is directly related to particle diameter
and was calibrated prior to deployment using polystyrene latex sphere calibration standards (PSL with 0.8,


0.9, 1.0, 1.3, 2.0, 3.0 µm diameter, Thermo Scientific Inc.). The WIBS optical size therefore refers to PSL
material with a real refractive index of 1.59. Healy et al. (2012) determined WIBS-4 counting efficiency by
aerosolizing standardized concentrations of PSL sphere of specific sizes (e.g. 0.3, 0.4, 0.56, 0.7, 0.9 and
1.3µm) and compared WIBS-4 total counts against PSL counts detected by the condensation particle
counter (CPC). Results show WIBS-4 possesses a 50% counting efficiency for 0.5µm particles and detects
100% of the PSL particles above 0.7µm when it is compared to the CPC counts. The 280nm and 370nm
pulsed Xenon flashtube UV lights in the WIBS cause the particles to autofluoresce (i.e., excite the
chromophores preexisting in the PBAP and do not rely on a fluorescent dye as done in FCM). Then,
fluorescent emissions are measured at three wavelength channels, which following the nomenclature of
Perring et al. (2015) are: (*i*) channel A ("FL1_280" in previous studies; Robinson et al., 2013), which refers
to the detected emission between 310-400nm after excitation at 280nm, (*ii*) channel B ("FL2_280" in
previous studies), which refers to the detected emission between 420-650nm after excitation at 280nm, and,
(*iii*) channel C ("FL2_370" in previous studies), which refers to the detected emission between 420-650nm
after excitation at 370nm. The resulting autofluorescence from 280nm excitation is affected by the presence
of tryptophan, tyrosine and phenylalanine aminoacids in the PBAP (Pöhlker et al., 2012). Similarly, the
resulting autofluorescence from the 370nm excitation is influenced by the presence of riboflavin and co-
enzyme Nicotinamide Adenine Dinucleotide Phosphate (NAD(P)H) within the cells.
Biological and non-biological particles can be discriminated by using a fluorescent intensity threshold;
here the threshold is determined with the Gabey et al. (2010) method and with modifications by Perring et
al. (2015) as follows. Particles with fluorescence intensities below the fluorescence threshold in all channels
are categorized as non-fluorescent (NON-FBAP). Particles that fluoresce above the threshold in only one
channel are named with a single letter (e.g. A, B or C); particles that fluoresce in two channels are named
with the two channel letters (e.g. AB, AC or BC), while particles that fluoresce in all channels are
categorized as type ABC. Furthermore, the total FBAP concentration is defined as the sum of the
concentration in the seven FBAP categories defined above. This approach was applied by Hernandez et al.,
(2016) to pure culture PBAP (bacteria, fungal spores, pollen) to study their correspondence to FBAP types;
bacteria tend to be detected by type A, and fungal spores and pollen by type AB and ABC. However,
bioaerosol classification is instrument-specific and particle size dependent (Hernandez et al., 2016; Savage
et al., 2017).
Several studies have used the Perring et al. (2015) FBAP categories to characterize PBAP in multiple
environments across the globe (Yue et al., 2017; Gosselin et al. 2016 ; Yu et al., 2016). Perring et al. (2015),
using a WIBS-4, studied atmospheric PBAP onboard a Skyship 600 aircraft operating between 300m and
1km above ground level at 10 geographic regions across the United States; the study concluded that type



AB (~30%) and ABC (~25%) is the most abundant of FBAP particles in the Southeastern US (East Texas
to Central Florida), and AB (~1.9 µm) and ABC (~2.6 µm) median sizes are characteristic of mold spores
(fungal spores of unknown amount of species predominant on humid and warm environments;
www.cdc.gov). In addition, FBAP concentrations in the Southeastern US range from $2\times10^4$ to $8\times10^4$ m$^{-3}$,
constituting 3-24% of the total supermicron particle number between 1 and 10µm diameter. In the
Southwestern US, Perring et al. (2015) shows AB and ABC types contribute less due to a higher relative
contribution by types B (~25%), BC (~20%) and C (~5%), and total FBAP constitute 5-10% of the total
supermicron particles. Furthermore, Perring et al. (2015) found the concentration of ABC type PBAP on
the surface and aloft did not vary throughout the Southeastern US. In the highly vegetated Rocky Mountains
Gosselin et al. (2016) found (using a WIBS-3) that ABC type particles always are a significant fraction of
FBAP (at least 20%) and are especially enhanced during rainy days (during or post-rain events) to ~ 65%
of the total FBAP, owing to the release of wet-ejected fungal spores following precipitation (Huffman et
al., 2013). However, during dry days, types BC and C increase their relative fraction to ~30% and ~40%,
respectively (Perring et al., 2015). Limited studies have looked closely at the FBAP categories in urban
environments. In Naijing, China, Yu et al. (2016) observed that types B (~45%), BC (~25%) and C (~15%)
dominate the FBAP concentrations during autumn. All FBAP types, except type C, correlated with black
carbon and PM$_{0.8}$ concentrations (particle mass with diameter below 0.8µm), suggesting a strong
interference by combustion sources; Type C PBAP ($6.6 \times 10^5 \pm 5.5\times 10^5$ m$^{-3}$) was considered more
representative of bioaerosols, although with unknown interference from abiotic particles. Similarly, Yue et
al. (2017) found a dominance of type B PBAP (~66% of total FBAP) during clean and polluted events in
wintertime Beijing, China; interestingly, the FBAP contribution to the total particle concentration is higher
during polluted events (13-24%) than during clean events (12-14%). FL1 type particles (sum of types AC,
ABC, AB and A) are more abundant in clean periods (~25%) than in polluted periods (10.1%), while the
fraction of type C FBAP is higher during polluted periods (~20%) than during clean periods (~5%).
**2.4 Location of sampling site and sampling frequency**

Bioaerosol sampling was conducted between April 7 and May 15, 2015 at the rooftop sampling

platform of the Ford Environmental Sciences and Technology (ES&T) building at the Georgia Institute of
Technology campus in Atlanta, GA. The site, which was located at the heart of a major urban environment,
is surrounded by dense forested areas in the southeastern USA: the Oconee National Forest (South East),
the Chattahoochee National Forest (North), and the Talladega National Forest (West). The WIBS was
operating continuously throughout the same period, sampling bioaerosol from a 15 ft. long and ¼ in. ID
conductive tubing inlet fixed 8 ft. above the sampling platform floor. The SpinCon II was placed in the
platform during sampling episodes with its inlet facing South. Three 4-hour samples per week were
collected with the Spincon II sampler over the 5-week period (4 h sampling between 10am and 5pm; Table





1). Meteorological data acquired from the same platform provided wind speed, wind direction, relative
humidity (RH), temperature, total hourly rain and UV radiation index with a 1min resolution.

**3. Data processing and Analysis**
**3.1 FCM data processing**
All blanks collected showed contamination levels that did not exceed 1% of the PBAP quantified in the
subsequent atmospheric samples. The 2-minute instrument blanks obtained after the CP and the HEPA filter
washes was $1.06\times10^3 \pm 7.37\times10^2$ mL$^{-1}$ and $9.22\times10^2 \pm 1.24\times10^2$ mL$^{-1}$, respectively, which are negligible
accumulations compared to the $2.55\times10^5 \pm 1.14 \times10^5$ mL$^{-1}$ average PBAP concentration quantified in the
atmospheric samples. The concentration of PBAP in the blanks was also confirmed with microscopy (not
shown). Based on this, we are confident that the CP protocol and procedure to replace the working fluids
ensured sterility of the biosampler before each sampling.
FCM analysis of the samples was carried out as follows. We obtain the fluorescence intensity (from
each of the 4 fluorescence detectors), forward scattering and side scattering intensity for all the particles
suspended in the samples. A gating procedure was used to determine the fluorescence levels associated
with detecting only particles containing SYTO-13 (hence, a PBAP) and background fluorescence from non-
stained particles. The procedure (Supplemental information, SI.2 and SI.3) consists of 3 steps: (a)
fluorescence threshold determination, (b) population gating, and, (c) biological/non-biological particle
discrimination in the population(s) within the threshold (e.g. LNA PBAP, Section 4.1). The fluorescence
threshold was determined using an atmospheric sample without SYTO-13 collected before each FCM
analysis, as a blank. Based on the fluorescence responses obtained, we determine the FL1-A fluorescence
intensity value for which 99.5% or 99.9% of the (unstained) particles of the blank autofluoresce below the
chosen value. This FL1-A intensity, called "fluorescence threshold", was determined for each sample
(supplementary information, Figure S2a and S2b). The determination of the fluorescence threshold
involved selecting the most conservative value that maximizes inclusion of biological particles and
minimizes the inclusion of non-biological particles, including those that may be subject to background
fluorescence or unspecific binding of SYTO-13 (Diaz et al., 2009; Müller et al., 2010). We found out that
threshold values for the 99.9% approach were substantially higher than 99.5% approach in multiple
sampling events and comparable to the fluorescence intensities observed for stained pure cultures (~$10^5$
units), which means that the 99.9% threshold values will miscount pure cultures as non-biological.
Consequently, we set the fluorescence threshold to the highest fluorescence intensity value observed by the
99.5% approach (41,839 units; supplementary information, Figure S2b), applied it to all collected samples;
henceforth named the 42k FL1-A threshold. The 42k threshold value aims to minimize any abiotic





interference as it maximizes biological particles quantification. A fixed value has been chosen and applied to all samples given that having a different threshold value for each sampling event may result in quantification biases as bioaerosols with strong autofluorescence (e.g. pollen, fungal spores) can increase the threshold value and affect PBAP quantification in the population(s) within the threshold. The BD Accuri C6 flow cytometer used for the analysis of the samples maintains constant pre-optimized photomultiplier voltages and amplifier gain settings. As a result, the fluorescence intensity of particles is consistent from day-to-day, and the fluorescence intensity of a specific biological particle population having the same metabolic state and physiological characteristics must not show day-to-day variability (www.bdbiosciences.com). Under the 42k threshold approach PBAP concentrations in the population(s) within the threshold (e.g. LNA, Section 4.1) can be overestimated by up to a 0.5%. Furthermore, FCM experiments conducted with unprocessed Arizona Test Dust (ATD) show that the FL1_A intensity distribution of SYTO-13 stained ATD particles is very similar to unstained ATD particles, and 100% of the SYTO-13 stained ATD particles stay below the 42k threshold (supplemental information, Figure S14a and S14b), supporting the 42k threshold effectiveness to filter out abiotic particles.

Once the FL1-A threshold was determined, plots of FL1-A vs. SSC-A and FL1-A vs. FSC-A are used to define clusters of bioparticles with fluorescence that exceed the FL1-A threshold and a characteristic optical size (obtained from the FSC-A intensity) or particle shape/complexity (obtained from the SSC-A intensity). FL1-A vs. SSC-A plots were used to define the populations of bioparticles for PBAP quantification as clusters using SSC-A parameter were more defined and showed better spatial resolution than using FSC-A parameter. The limits of each population were also determined with Flow Jo (www.flowjo.com), using 2% contour plots (supplemental information; Figure S3) generated by equal probability contouring (i.e., 50 contour levels so that the same number of cells fall between each pair of contour lines). Populations above the FL1-A threshold value (41,839 FL1-A units) were considered biological (Section 4.1; e.g. HNA); the particles in the population within the threshold value (Section 4.1; e.g. LNA) having a FL1-A intensity greater than 41,839 units were counted as biological to determine the PBAP counts in the population. The total PBAP counts were considered as all particles counts having FL1-A fluorescence intensity above the determined threshold value minus the 15µm beads internal standard having FL1-A fluorescence intensity above the determined threshold value. The 15µm beads of known concentration and particle size allows for calibrating the optical size (supporting information, SI.7) of the bioparticles, as well as their concentration and departure from sphericity. The 15µm beads population showed fluorescence intensities comparable to the determined fluorescence threshold after been stained with SYTO-13 as it is known that molecular stains can be adsorbed on the surface of polystyrene beads (Eckenrode et al., 2005; Rödiger et al., 2011). The relatively high fluorescence intensity of the 15µm beads show populations within the threshold value (e.g. LNA, Section 4.1) cannot be rule out as being affected



by unspecific staining of abiotic particles. However, populations above the threshold value (e.g. HNA,
Section 4.1) should not be affected by such abiotic interferences.

**3.2 WIBS data processing**

15-minute average total aerosol and FBAP size distributions were obtained from the WIBS. FBAP was
distinguished from the total aerosol using the Gabey et al. (2010) "trigger threshold" approach, which is
applied as follows. First, the average "electronic fluorescence noise" and its standard deviation is
determined for each channel (A, B, C) performing the Force Trigger (FT) calibration which consist to
operate the WIBS without flowing air through the system. The FT calibration, carried out every 24hr, is
critical for determining the lowest particle autofluorescence levels that robustly exceeds instrument
electronic noise. FT calibrations measured the particle-free air background autofluorescence in the three
WIBS channels (e.g. A, B, C), and measurements recorded the fluorescence intensity for 500 excitation
flash events (Ziemba et al., 2016; Tropak and Schnaiter, 2013; Gabey et al., 2010). The threshold for each
detector is then equal to the average fluorescence plus 2.5 times its standard deviation; particles with
fluorescence intensities above this threshold value are classified as FBAP. Then, Perring et al. (2015)
approach (Section 2.3) is applied to determine the combination of thresholds that provide the maximum
concentration of PBAP and minimal interference from abiotic particles, which still remains an area of active
research. WIBS-3 and WIBS-4 models have been actively studied to determine which channel best detect
bioaerosols and to cluster different types of PBAP (Robinson et al., 2013; Crawford et al., 2014; Gabey et
al., 2010). Both models use filtered xenon flash lamps to excite particles at 280nm and 370nm wavelengths
and detect PBAP autofluorescence in two regimes (For WIBS-3, FL1: 320-600nm and FL2: 410-600nm;
For WIBS4, FL1: 310-400nm and FL2; 420-650nm). Three separate fluorescence channels for each model:
(i) channel A: detection in FL1 following 280nm excitation, (ii) channel B: detection in FL2 following
280nm excitation and (iii) channel C: detection in FL2 following 370 nm excitation, are then available for
FBAP determination. The main difference between WIBS models is that the fluorescence detection regimes
overlap in channels A and B for the WIBS-3, but not for the WIBS-4. WIBS-3 FBAP quantification cannot
be compared directly with WIBS-4 due to channel A and B overlap, but FBAP detection in all channels
have been consistent between both models (Robinson et al., 2013). WIBS-4 contains two switchable gain
settings (e.g. high gain (HG), low gain (LG)), allowing it to measure 0.5μm to 12μm particles in HG and
3μm to 31μm particles in LG setting. On the other hand, the second generation of the WIBS-4, named
WIBS-4A, maintains single gain settings and evaluates particles between 0.5μm and 20μm (Fennelly et al.,

2017).

Gabey et al. (2011) concluded, using a WIBS-3, that channel C was most efficient in quantifying FBAP
either in the Borneo tropical forest or in the urban environment of Manchester, UK. Healy et al. (2014)
found higher channel A FBAP concentration in Killarney, Ireland using WIBS-4. Pure culture experiments



with WIBS-4 have shown high detection efficiency of channel A toward *Pseudomona syringae* bacteria
(Tropak et al., 2013). Hernandez et al. (2016) used WIBS-4 to test the intrinsic fluorescence fingerprints of
29 fungi, 13 pollen and 15 bacteria species and suggested channel A is most suitable for discriminating
bacteria and fungi, channel C is most suitable for pollen and channel B can be influenced by abiotic
particles. In addition, among FBAP categories (Perring et al., 2015) bacteria is mainly detected as type A,
fungal spores shown multiple fluorescence types (e.g. A, AB, BC and ABC) and pollen is mainly detected
as type BC and ABC.  However, PBAP detection effectiveness by specific channels varies considerably
between instruments, which suggests a thorough calibration may be necessary. Furthermore, Savage et al.
(2017) used WIBS-4A to show FBAP fluorescence also varies with particle size, especially for pollen and
fungal spores and proposed pathways of change by which particles may transition from type A or type B to
type ABC as they increase size. FBAP type variation with particle size is important to consider as the
approach of Perring et al. (2015) is used to better understand what FBAP type is best detected (e.g. bacteria,
fungal spores, pollen).

In this study, thresholds for each channel were determined daily, and the total particle concentration,

FBAP types (e.g. A, B, C, AB, BC, AC, ABC) concentrations and the total FBAP concentration (sum of
the seven FBAP types) were used. From the data, 4h-averaged size distributions (using 15-minute average
data) were generated for the total particles and all FBAP types in the 1-10µm range during the time SpinCon
II run. Subsequently, WIBS overall sampling efficiency (aspiration efficiency + transport efficiency) was
calculated using the Particle Losses Calculator (Von der Weiden et al., 2009) and applied to the 1-10µm
size distributions for the sampling characteristics in our setup (15ft. sampling line with ¼ in. ID and 2.3 L
min$^{-1}$ flow rate; Figure S4a). The sampling efficiency was calculated to be 67% for 5µm particles, with
larger losses as size increased to 10µm. (supplemental information, FigureS4b). FCM and WIBS total
particles and PBAP comparison was constrained to the 1 to 5µm range being the size overlap of both
techniques. Also, the fractional composition of FBAP (based on number concentrations) was calculated to
characterize its daily variability (Section 4.2), and compared against the daily variability of PBAP from the
FCM analysis (Section 4.4).
**4. Results and Discussion**
**4.1 FCM biopopulation identification and quantification**

When the FCM results are plotted in terms of FL1-A intensity versus SSC-A intensity, four populations

(Figure 2) emerge above the threshold gating process: low nucleic acid (LNA) particles, high nucleic acid
(HNA) particles, pollen and the 15µm internal standard beads. EPM and SEM pictures (Supplementary
Figures S5, S6, and S7) confirm the presence of these heterogeneous populations. Previously, LNA and
HNA populations were identified in FCM of aquatic samples with the use of the SYTO-13, SYBR green



and DAPI nucleic acid stains (Wang Y. et al 2010; Bouvier, T. et al 2007; Lebaron, P. et al 2001);
corresponding populations in atmospheric PBAP have not been identified before. Below we focus on each
population to further identify them as pollen, fungal spores, bacteria and other fragments.
The HNA size distributions are dominated by 3-5µm particles (mean diameter: 4.15 ± 0.06 µm;
Supplemental Information, Figure S10) and the total concentration strongly correlated with RH. HNA were
virtually non-existent during extended dry periods (days with average RH < 70% during sampling, e.g. 4/9,
4/22 and 5/15) and well defined during periods of high humidity, especially after rain events (days with
average RH > 70% and T > 18 °C during sampling episode; e.g. 4/7, 4/14, 4/15). Both these characteristics
suggest that HNA particles correspond to fungal spores (e.g., from the Ascospores and Basidiospores genus;
Oliveira et al., 2000; Li and Kendrick, 1995). The LNA size distributions are dominated by 2-4 µm particles
(mean diameter: 2.99 ± 0.06µm; Supplemental Information, Table S1) and dominated Atlanta PBAP
composition during dry days. The LNA population shows SYTO-13 fluorescence intensities that are about
one order of magnitude lower than the HNA population. The observed particle sizes are within the size
range of airborne bacteria (Després et al., 2012) and the LNA population may represent single or
agglomerated bacterial cells. However, it is clear that heterogeneous populations will probably contain
multiple types of microorganisms and that may be the case in the LNA population.
It is known that pollen may burst into tiny fragments when is suspended in water (e.g., Augustin et al.,
2012; Taylor et al., 2007). Therefore, pollen may fragment during sampling and processing of samples in
the FCM, increasing the concentration of LNA particles and biasing concentrations. FCM applied to
ragweed pollen suggests a 1:2 pollen-to-pollen fragments concentration ratio (Supplementary information,
Table S2). Also, calculations based upon FCM-derived ragweed pollen and pollen fragments concentrations
during this study (considering the total pollen mass added to the sample, 15µm mean diameter previously
determined by Lin et al. (2013) and unit density) suggest approximately 67% of the ragweed pollen grains
were intact after hydration and that each fragmented grain generates ~5 pollen fragments; in agreement
with Bacsi et al. (2006), 35% of ragweed pollen fragments upon hydration. Overall, ragweed pollen results
suggest FCM experiments do not have a considerable impact in pollen fragmentation and that pollen
fragmentation will have a negligible effect on LNA concentrations. Ragweed pollen is one of the most
abundant wind-driven pollen species in the United States and its emission peaks during fall, but can be also
present during late spring and summer. It is representative of the pollen species we see in the Atlanta area
(Darrow et al., 2012) and results suggest pollen fragmentation would not generate a substantial amount of
fragments. The low collection efficiency of SpinCon toward large particles (<14% for diameters above
5µm) and that pollen concentrations in our samples are generally two orders of magnitude lower than LNA
concentrations suggest a negligible effect of pollen fragments in LNA biological particle quantification.





Also, EPM results showed intact pollen and limited amounts of small debris among the particles identified
in the atmospheric samples collected for this study. Particles with fluorescence intensities above the FL1-
A threshold value were counted as biological, giving us the PBAP counts within the LNA population and
will be referred henceforth as the "bioLNA" population (Figure 2).

A population of strongly fluorescing and very large particles (10-20µm, avg. average geometric mean

diameter $12.3 \pm 1.7\mu m$) was identified (Figure 2). This population also strongly autofluoresces in the FCM
when SYTO-13 was not added to the sample (SI.7, Figure S11). All together this indicates a population of
pollen particles, as they are known to contain cell wall compounds (i.e., phenolic compounds, carotenoid
pigments, Phenylcoumarin) that fluoresce more strongly than the proteins and cytosolic compounds
responsible for bacteria/fungi autofluorescence (Pöhlker et al., 2012; Hill et al., 2009; Pöhlker et al., 2013).
The pollen population was not well-defined during all sampling events; whenever present, pollen was
characterized by concentrations ($\sim 10^2 \, m^{-3}$) consistent with reported values (Despres et al., 2012), which are
also much lower than bioLNA and HNA concentrations. As a result, pollen population was systematically
gated using a perfect square between $10^6$ and $10^8$ intensity units in the FL1-A vs. SSC-A plot for each
atmospheric sample. bioLNA, HNA and pollen counts, acquired by the 42k threshold approach were used
to calculate liquid-based ($mL^{-1}$ of sample solution) and air-based ($m^{-3}$ of air) concentrations for each
bioaerosol population as detailed in the Supplemental Information. The total PBAP concentration on each
sample consisted of all non-bead particles above the 42k fluorescence threshold given that a non-negligible
biological particle concentration was not constrained in the gated populations. Even though the 2% contour
plots effectively allowed population gating, $16.5 \pm 7.3\%$ of the total PBAP are not attributed the identified
populations. The biological particles not constrained by gating, henceforth named as the "unclassified"
bioparticles, showed the highest concentrations when both HNA and LNA populations are densely
populated (4/16, 4/28 and 5/14; Figure 5). The lowest concentrations were observed when just the LNA
population is identified (4/9, 4/22, 5/15; Figure 5) and when the LNA and HNA populations are identified
after the rain event on 4/14. The observed behavior shows that the unclassified bioparticle concentrations
is linked to the heterogeneity of the biological populations and the concentration of the gated populations
(e.g. HNA, LNA and Pollen). The "unclassified" bioparticles concentration ranges from $8.1 \times 10^2 \, m^{-3}$ to
$1.3 \times 10^4 \, m^{-3}$ (avg. $4.2 \times 10^3 \pm 3.3 \times 10^3$) and they are not constrained to a specific size range. In addition,
we must note that additional concentration corrections are required owing to the sampling efficiency of the
SpinCon II, but will be considered in sections 4.3 and 4.4.

Before SpinCon II sampling efficiency corrections are applied, FCM total particle concentrations range

from $2.6 \times 10^4 \, m^{-3}$ to $2.9 \times 10^5 \, m^{-3}$, with increasing concentrations toward the end of the sampling period.
In addition, total PBAP concentration averaged $2.4 \times 10^4 \pm 1.1 \times 10^4 \, m^{-3}$ (coefficient of variation, CV, 13%;
defined as the standard deviation over a triplicate FCM measurements over the average concentration).



BioLNA ranged between $6.8\times10^2$ and $2.9\times10^4$ m$^{-3}$ (average: $1.1\times10^4$ m$^{-3}$; CV: 20%), HNA(fungal spores)
between $4.7\times10^3$ and $1.9\times10^4$ m$^{-3}$ (average: $1.1\times10^4$ m$^{-3}$ ; CV: 15%) when above the detection limit (n=12),
and pollen from $1.3\times10^2$ to $1.2\times10^3$ m$^{-3}$ (average: $3.6\times10^2$ m$^{-3}$; CV: 21%). These concentration levels are
consistent with microscopy-based studies in urban environments for bacteria (e.g., $1.7\times10^4 \pm 1.3\times10^4$ m$^{-3}$
in springtime Birmingham, UK; (Harrison et al., 2005); fungal spores ($1.8\times10^4 \pm 1.1\times10^4$ m$^{-3}$ in Vienna,
Austria between April-June; Bauer et al., 2008); and pollen (between $5.69\times10^2$ m$^{-3}$ to $6.144\times10^3$ m$^{-3}$ in
Medellin, Colombia; Guarín et al., 2015). Also, additional experiments performed in September 2015,
described in Figure S7 of the supplemental information (supplemental information, SI.6), showed that EPM
and FCM-based quantifications agree within an order of magnitude. This is consistent with Lange et al.
(1997), whom also found that FCM gives higher quantifications than EPM microscopy when studying *P.*
*aeruginosa* pure cultures and airborne bacteria collected from a swine confinement building in Iowa, USA.
To better understand SYTO-13 fluorescence intensity differences between the identified (e.g. bioLNA,
HNA and pollen) populations in the atmospheric samples and their metabolic/stress state, FCM experiments
were conducted with air-isolated bacteria (F8 strain; De Leon Rodriguez, 2015), ragweed pollen and yeast
(*S. cerevisiae*; Y55 strain) mixtures to compare the SYTO-13 fluorescence intensity and the scattering
properties of the pure cultures to those seen in the atmospheric samples. Pure cultures and atmospheric
samples are summarized in Tables S3, S4 (supplementary information; FCM pure culture experiments)
respectively. The bioLNA population showed SYTO-13 fluorescence intensity up two orders of magnitude
lower than F8 bacteria. HNA (fungal spores) population showed an order of magnitude lower SYTO-13
fluorescence intensity than Y55 HNA yeast, and, within the same magnitude for the LNA Y55 yeast. The
HNA and LNA yeast populations in the pure culture experiments (Figure S13a) have one order of
magnitude difference in FL1-A fluorescence intensity and may represent yeast populations with different
metabolic states. Atmospheric and Ragweed pollen populations had similar SYTO-13 fluorescence
intensities and Figure S13c shows pollen fluorescence intensity may go up to $10^8$. The lower SYTO-13
fluorescence intensity of the atmospheric populations may be related to genetic material degradation from
exposure to atmospheric stressors; depending on the physiological characteristics of each population (Zhen
et al., 2013; Amato et al., 2015). Our results also agree with Guindulian et al. (1997), showing that *E.coli*
overnight cultures have higher SYTO-13 fluorescence intensity than starved *E.coli* population. Overall,
FCM pure culture results suggest microbes starve in the atmosphere, leading to a possible reduction or
leakage of the amount genetic material enclosed within each cell. Sampling can also stress cells, even
disrupt the wall/membrane of the cell and lead to genetic material leakage (Zhen et al., 2013).
Pollen, HNA (fungal spores) and bioLNA atmospheric populations showed different SYTO-13
fluorescence intensities. Pollen showed the highest fluorescence intensity, followed by the HNA and





bioLNA (fraction of LNA above threshold; Figure 2) populations, respectively (Figure 2; Table S4).
Guindulian et al. (1997) FCM results with starved bacterioplankton from seawater samples treated with
DNASe/RNase showed SYTO-13 fluorescence intensity can be related to the DNA content of starved
bacterioplankton due to the low amount of RNA enclosed in starved cells. Taking in consideration our
results and previous studies, we can suggest that Pollen, bioLNA and HNA populations in the atmospheric
samples are differed by their DNA content, which can in part explain SYTO-13 fluorescence intensity
difference between them. Future work is needs to further study this.

## 4.2 WIBS total concentration and FBAP daily variability

WIBS-4A collected data continuously throughout the period; for comparison against the SpinCon
II 4h liquid batch samples, WIBS data was averaged to the SpinCon II sampling times (Table 1). WIBS
total particle concentration (1-5µm diameter) ranged from $2.0 \times 10^5$ to $1.0 \times 10^6$ m$^{-3}$ in agreement with
observed particle concentrations in previously studied urban environments during Spring/Summer months
like Helsinki, Finland (UV-APS avg. $1.6 \times 10^5$ m$^{-3}$; Saari et al., 2015) and Karlsruhe, Germany (WIBS-4
avg. $6.9 \times 10^5$ m$^{-3}$; Tropak and Schnaiter et al., 2013). 4h average total particles concentrations in Figure 3a
show particle concentrations declined during rain episodes (during or post-rain: e.g. 4/15, 4/16, 4/28, 4/29,
4/30) as wet removal of PBAP is most efficient. However, during dry (no rain) episodes total particle
concentrations built up in the atmosphere. To better understand the day-to-day variability of different FBAP
types, the seven Perring et al. (2015) FBAP categories (e.g. Type A, B, C, AB, AC, BC and ABC) were
studied plus the NON-FBAP type constituting particles that do not fluoresce in any channel (e.g. channel
A, B, C). NON-FBAP concentrations are one order of magnitude higher than FBAP concentrations, and
NON-FBAP, hence traced WIBS total particles throughout all sampling events (Figure 3a). Total FBAP
concentrations also show similar behavior to the total particle concentration (Figure 3a) and it suggests non-
biological particles can be biasing the total FBAP concentration. The variability of the total FBAP
concentration is mainly linked to type A and type B concentrations as overall they constitute the two largest
fractions to the total FBAP concentration (Figure 3b), and both FBAP types have previously misidentified
non-biological particles as FBAP (Tropak and Schnaiter et al., 2013; Yu et al., 2016).  As a result, our study
considers the total FBAP concentration as the upper limit, and ABC type concentration as the lower limit
of FBAP concentration in Metro, Atlanta. Type B dominates the FBAP fractional composition (Figure 3b),
which has been linked to possible non-biological interferences from black carbon (Yu et al., 2016) and
polycyclic aromatic hydrocarbons (PAHs) emitted from combustion sources.  Total FBAP fraction ranges
from 16% and 43%, and ABC fraction ranges from 1.3% and 9.2% of the total particles in the 1 to 5µm
size range. ABC type fractions and ABC type concentrations are within the values observed by Tropak and



Schnaiter (2013) using WIBS-4 in Karlsruhe, Germany; averaging $2.9 \times 10^4 \, \text{m}^{-3}$ (when considering the sum
of AC and ABC types) and constituting 7% of total coarse mode particles (0.8μm-16μm).

ABC type concentrations show an interesting variability throughout the 15 sampling events, as

ABC reaches its maximum concentration on 4/14, on a warm and humid day after a rain event, concurrently
when the FCM HNA population also reaches its highest concentration – strongly suggesting ABC particles
are fungal spores. (Figure 3a, Table 1). Recently, Gosselin et al. (2016) used WIBS-3 in the Rocky
Mountains, Colorado showing ABC type fractional composition enhances after rain events to dominate the
total FBAP composition and the enhancement is correlated to mannitol and arabitol concentrations (fungal
spore tracers), which have been previously linked to Ascomycota and Basidiomycota spores emitted by the
wet-ejection mechanism (Elbert et al., 2007). In addition, ABC type constitute a considerable fraction
(~20%) of total FBAP during dry days in the Rocky Mountains possible because such highly vegetative
environments maintain a high background of fungal spores (Huffman et al., 2013). However, urban
environments like Metro Atlanta are not necessary dominated by fungal spores and its FBAP composition
will be affected by the biological sources close to city (e.g. forests), local emissions and meteorology. The
overall FBAP composition in metro Atlanta (Figure 3b) is dominated by type B (avg. fraction: $33 \pm 9$%),
type A (avg. fraction: $22 \pm 5$%) and type AB (avg. fraction: $22 \pm 5$%) particles. Type ABC constitute $12 \pm$
6% of the total FBAP and it reaches 30% on 4/14, comparable to values observed by Gosselin et al., 2016
in the Rocky Mountains. The dominance of type B particles has been observed in the polluted atmosphere
of Nanjing, China using WIBS-4A were type B constituted ~ 45% of the total PBAP and type B (~ $2 \times 10^6$
$\text{m}^{-3}$ ) concentrations were up to two orders of magnitude higher than type A concentrations (~$5 \times 10^4 \, \text{m}^{-3}$ )
suggesting a high likelihood of interference from abiotic particle sources. However, Metro-Atlanta shows
much lower total particle concentrations than Nanjing, China (~$10^7 \, \text{m}^{-3}$) and type A and type B
concentrations are within the same order of magnitude. Furthermore, Perring et al. (2015) have shown type
B particles constitute a considerable fraction of the total supermicron particles across the United States,
being ~15% and ~25% over (altitude >100m) the Southeastern US and Southwestern US, respectively.
Total particle and NON-FBAP size distributions in Figure 3c peaked at ~1μm. Similarly, types A, B, AB
size distributions (Figure 3d) peaked close to 1μm showing that interferences by non-biological particles
cannot be rule out. However, ABC type size distribution (red line, Figure 3d) is dominated by 3-5μm
particles and ABC type particles may have come from a different source to other FBAP types as they get
enhanced after rain events (e.g. 4/14; Table 1). Yu et al. (2016) also observed 4-6μm ABC type particles
in the highly polluted Nanjing, China, but ABC type bimodal size distributions showed a peak between 1-
2μm and a second peak between 4-6μm. In addition, ABC type number fractions in Nanjing, China
correlated to black carbon mass fractions suggesting a considerable influence by combustion related





particles and no rain events occurred during the sampling period. The difference between Metro Atlanta
and Nanjing, China ABC type size distributions suggest ABC type is not influenced by combustion related
particles in Metro Atlanta. Overall, results show FBAP concentration (1-5µm) ranges from $10^4$ -$10^5$ m$^{-3}$ in
metro Atlanta and wet-ejected fungal spores concentration, detected by ABC type, can constitute up to 30%
of the FBAP (1-5 µm) after rain events.

## 4.3 Correlation of HNA population with ABC type

A quantitative comparison between WIBS-4A total particle and FCM total particle concentrations
was subsequently performed and we focused the analysis to the 1 to 5µm size range as SpinCon sampling
efficiency is reduced significantly above 5µm (≤14%; Kesavan et al., 2015). WIBS-4A and FCM total
particle concentrations differed by about one order of magnitude (for optical diameter, $d_o$, greater than
1.5µm) and particle concentration difference increased for particles with $d_o < 1.5$ µm as shown in the size
distribution (geometrically averaged across the 15 SpinCon II sampling events) in Figure 4a. The largest
difference between WIBS-4A and uncorrected FCM size distributions seems to be related to SpinCon II
having a cutoff size close to 1µm, reducing significantly its sampling efficiency. Even with the observed
difference in the magnitude of the concentrations between the two techniques, ABC type and HNA
concentrations traced throughout all the sampling events and are highly correlated ($R^2 = 0.40$; Figure 4b)
and showed similar size distributions in the 1 to 5µm range as shown in Figure S12a. HNA and ABC type
were both dominated by 3-5µm particles and its seems both are detecting the same type of biological
particles. In addition, AB type showed a weak correlation with HNA concentrations ($R^2 = 0.17$), but their
size distributions differed as type AB peaks close to ~1µm (Figure 3d). ABC is the only FBAP type showing
a considerable correlation to the HNA population, and bioLNA population is not correlated with any FBAP
type. Overall, ABC type and HNA correlation is an important step forward to better understand the
effectiveness of WIBS-4A FBAP categories to provide speciated PBAP concentrations in urban areas. ABC
type particles have shown substantial concentrations ($10^4$-$10^5$ m$^{-3}$; Perring et al., 2015; Ziemba et al., 2016)
across the US. The highest ABC fraction of the total FBAP was observed in Panhandle, Florida during an
airborne study among multiple environments studied using WIBS-4A to sample from the California coast
to central Florida, suggesting ABC type particles are ubiquitous in the US (Perring et al., 2015). Previous
studies (Healy et al., 2014, Huffman et al., 2013) have shown correlations between LIF technology (e.g.
WIBS-4 and UV-APS) fluorescence channels and fungal spores number concentrations, especially during
fungal spores invigoration after rain events. Healy et al. (2014) used WIBS-4 in Killarney National Park,
Ireland (e.g. high vegetative rural area) finding correlations between channel B (FL2; $R^2 = 0.29$) and channel
C (FL3; $R^2 = 0.38$) concentrations and fungal spores concentrations (collected by Sporewatch impactor and
quantified by microscopy). However, now for the first time FCM HNA population have shown a correlation
with WIBS-4A ABC type and suggests ABC type category detects well actively ejected fungal spores in





Metro Atlanta (e.g. urban area). In addition, recent WIBS-4A experiments using pure cultures have shown
ABC type detects well several fungal spores (e.g. Aspergillus Versicolor & Botrytis spp.) and small pollen
grains, but detection may vary across instruments (Hernandez et al., 2016).

FCM concentrations were corrected based on correction factors (CF) calculated upon the

comparison of ABC and HNA size distributions (1 to 5µm) for each sampling event given (1) ABC type
and HNA population similar size distributions and number concentrations (1 to 5µm) correlation, and, (2)
WIBS-4A provides us representative concentrations of airborne particle concentrations in Metro Atlanta
after sampling losses being corrected (Section 3.2). Concentration correction factors were determined for
each sampling episode by taking the quotient of ABC type to HNA concentrations over the 1-5µm size
range. The resulting size-dependent correction factor (Figure S12b) was then applied to the FCM size
distributions, giving the "corrected FCM" bioaerosol data (between 1 and 5 µm). Figure 4a shows that the
corrected FCM total particle average size distribution traces WIBS-4A size distribution, allowing us to
correct for SpinCon II low collection efficiency and to better constrain the magnitude of FCM
concentrations. Our approach to calculate the estimated collection efficiency (ECE) considers all the
processes that affect the concentration of PBAP, from collection to final quantification in the FCM. Figure
S12b compares Kesavan et al. (2015) collection efficiencies determined for SpinCon I and the estimated
collection efficiency calculated upon the CF calculation (ECE = 1/CF) and shows the ECE of the SpinCon
II is lower that Kesavan et al. (2015) below 3µm and performs better for particles above 3µm, but above
3µm Kesavan et al (2015) collection efficiency is within the uncertainty of our calculations. Our lower ECE
values (Figure S12b) for particles below 3µm can be related to SpinCon sampling time as Kesavan et al.
(2015) experiment were conducted in a short period of time (e.g. 10-15 min) and ours took place for 4 hr.
The main mechanisms leading to below 3µm particle losses could be their re-arosolization over time being
lost through the blower exhaust of the SpinCon II (Figure 1). Also, coagulation of small particles over time
can not be rule out, but future work is needs to study it.

**4.4 PBAP populations after collection/detection corrections**

After correction through the application of the ABC correction factors, FCM total particle

concentrations (1 to 5µm avg.: $5.5 \times 10^5 \pm 5.1 \times 10^5$ m$^{-3}$; Figure 5a) are within the same order of magnitude
as WIBS-4A concentrations (1 to 5µm avg.: $5.4 \times 10^5 \pm 2.9 \times 10^5$ m$^{-3}$; Figure 3a), and continue to exhibit
substantial variability. The HNA (e.g. fungal spores) population showed a substantial invigoration during
three sampling events (4/7, 4/14, 4/15; Figure 5a and 5b). To better understand the role of meteorology on
PBAP composition, the PBAP samples were classified into four regimes based on the average diurnal
relative humidity and ambient temperature, with T= 18 °C (65 °F) to differentiate between warm and cold





734 days, and, RH = 70% to differentiate between humid and dry days. The temperature and RH threshold

735 values were chosen based on the observations and understanding that a combination of temperature and RH

736 within these threshold values can significantly impact bioaerosol composition. For instance, humid and

737 warm conditions  may lead to the invigoration of fungal spores by wet ejection from plants (Ingold, 1971),

738 on contrary, PBAP will get stressed when exposed to warm and dry conditions. The sampling times, RH,

739 ambient temperature and meteorological categories of each SpinCon II sample is presented in Table 1.

740 Humid and warm days (4/7, 4/14 and 4/15; light green shaded areas in Figure 5a) were characterized

741 by well-defined HNA and bioLNA populations. These sampling episodes had the highest average HNA

742 (fungal spore) concentration ($4.0\times 10^4 \pm 1.3\times10^4$ m$^{-3}$) among the four meteorological regimes and during

743 these sampling events HNA constituted $\geq 77$ % of the total PBAP. Among the humid and warm days (Figure

744 5a and 5b), average bioLNA, HNA and "unclassified" bioaerosol compositions were 6.1%, 84.0% and

745 9.9%, respectively of the total PBAP number. Also, the humid and warm days occurred after rain events,

746 which can be linked directly to the strong fungal spore invigoration (Huffman et al., 2013). Before sampling,

747 early morning precipitation occurred during 4/14 and 4/15, as well as during the night of 4/6. Precipitation

748 did not occur during sampling in any of the humid and warm days. The FCM results (Figure S15a-c) that

749 display the PBAP population between 4/7 and 4/9 show a disappearance of the (HNA) fungal spore

750 population during the transition from a "humid and warm" day (4/7) to a "dry and warm" day (4/9). Figure

751 5b shows how the HNA contribution to the total PBAP goes down on 4/8 when RH decreases and is

752 undetected on 4/9. Furthermore, Figure 6a-c shows FL1 vs. SSC-A plots for 4/14 to 4/16 consecutive

753 sampling periods, where a marked increase in the bioLNA concentration from 4/15 to 4/16 goes together

754 with a striking decrease in the HNA concentration. HNA fraction went down from 92.0.5% to 34.1% of the

755 total PBAP and bioLNA concentration went up from $3.8\times10^3$ m$^{-3}$ to $2.9\times10^4$ m$^{-3}$. Humid and Warm days

756 had the lowest averaged PBAP concentration ($4.6 \times 10^4 \pm 9.8 \times 10^3$ m$^{-3}$ in the 1 to 5µm range) among the

757 four meteorological regimes, a possible effect of the bioaerosols being lost by wet scavenging, resulting in

758 the enhancement of fungal spore contribution to the total PBAP number concentrations. The unclassified

759 biological particles concentration also showed its lowest contribution ($2.9 \times 10^3$ m$^{-3}$; 9.9%) to the total

760 PBAP number concentration during these events, when the HNA and LNA populations are best identified

761 by the 2% contour plots.

762 Cold and humid days (4/16 and 4/29; light yellow shaded areas in Figure 5a) also showed well-defined

763 HNA population, and HNA contributed on average to $29.5 \pm 6.5$ % of the total PBAP concentration (1 to

764 5µm). On 4/16 drizzling took place by the end of the sampling period, but no accumulated rainfall was

765 measured by the meteorological station. However, on 4/29, accumulated rainfall averaged 0.04in. from

766 11:55 AM to 2:20 PM (Figure S21). The similar HNA concentration between "Humid and Warm" and



"Humid and Cold" days seen in Figure 5a and the lower contribution of HNA to the total PBAP during the
"Humid and Cold" days may be linked to previously suggested bacteria emissions by droplet soil impaction
during rain events (Joung et al., 2017). Bacteria emission by soil impaction can increases airborne bioLNA
concentration and HNA (fungal spores) will have a lower contribution to the total PBAP even when the
fungal spore concentration is high during rain events. Both cold and humid days showed a considerable
difference in bioLNA contributions to the total PBAP concentration. On 4/16 and 4/29 bioLNA constituted
45.2% and 65.3% of the total PBAP concentration, respectively (Figure 5b). The difference in the bioLNA
contribution to the total PBAP can be linked to the intensity of precipitation, as it shapes the composition
(e.g. size and types) of microbes suspended in the atmosphere during the different stages of a rainfall (e.g.
before, on set, during and after a rainfall; Yue et al., 2016).
Six of the fifteen sampling days were classified as warm and dry (4/8, 4/9, 4/22, 5/13, 5/14, 5/15; light
orange shaded areas in Figure 5a) and it did not rain before or during any of these days (Table 1). During
warm and dry days, HNA had the lowest averaged concentration ($8.7 \times 10^3 \pm 1.2 \times 10^4$ m$^{-3}$) among the four
meteorological categories. In addition, during three dry and warm days (4/9, 4/22 and 5/15) the HNA
population was undetected. This behavior can be related to the fact that high RH drives fungal spore
emissions by wet ejection, but soil wetness could also affect emissions because the HNA population was
detected in other warm and dry days with comparable RH (Huffman et al., 2013; Gosselin et al., 2016). The
air mass trajectories reaching Atlanta during each sampling event could also affect the biological particles
composition. For example, on 4/22, when the HNA was undetected, the 500m and 100m 72 h backward air
mass trajectories reaching Atlanta came from the NW (US/Canada border) at high altitudes and do not
spend more than 24h near surface. This air mass could affect bioaerosol composition with minimal
influence from local bioaerosol emissions. However, the enhancement or the depletion of the HNA
population have not been linked to specific air masses trajectories. Besides meteorology, two main
hypotheses could explain the observed behavior in the HNA population, previously stated by Bouvier et
al., 2007 to understand HNA and LNA populations in aquatic environments, but also applicable to airborne
microorganisms. First, microbes might begin in the HNA population upon aerosolization and then move to
the LNA upon death or inactivity. Second, the HNA and LNA populations may contain completely different
microbial taxa and have different organisms in each population. If the first hypothesis occurs, we expect to
see a covariance of the HNA and LNA FCM parameters (e.g. FSC-A, SSC-A and FL1-A intensities), and
observe a gradual decrease in the FL1-A intensity of the HNA population to the FL1-A values observed by
particles in the LNA population, which is not seen. Although our results suggest the HNA and LNA are
two distinctive populations, further studies will have to take place to sort and directly study the DNA
sequences of each population in order to prove the second hypothesis. HNA population behavior may also
consist of a combination of both hypotheses. Overall, warm and dry days prevail during springtime in



Atlanta and bioLNA contribution (avg.: $3.4\times10^4 \pm 2.5 \times10^4$ m$^{-3}$) may represent the bioaerosol background
of Atlanta.

Four of the fifteen sampling days (4/21, 4/23, 4/28 and 4/30; light blue shaded areas in Figure 5a) were

characterized by cold and dry conditions (Table 1). PBAP were dominated by bioLNA during these events,
as can see in Figure 7a-c, where LNA population are the dominant contributors to PBAP number. HNA
population was diminished in Figure 7a (4/21) & Figure 7c (4/23) during cold and dry days and disappeared
in Figure 7b during a warm and dry day. Overall, HNA was detected during cold and dry days, but showed
lower contributions to the total PBAP number concentration than humid days. Among cold and dry days,
the PBAP population (1 to 5 µm) was composed on average of $72.6 \pm 10.1\%$ bioLNA and $16.5 \pm 8.2\%$
HNA. Cold and dry days had on average the highest bioLNA ($5.3\times10^4 \pm 1.8\times10^4$ m$^{-3}$) and total PBAP
($7.3\times10^4 \pm 2.0\times10^4$ m$^{-3}$) number concentrations (1 to 5µm) among the four meteorological categories,
reaching the PBAP maximum concentration on 4/23 (Figure 5a).

**4.5 PBAP day-to-day variability in Metro Atlanta: FCM vs. WIBS**

Although WIBS and FCM possess different methodologies, they show similar trends providing a

good understanding of the daily variability of PBAP in Metro Atlanta. FCM PBAP fraction (1 to 5µm)
ranges from 3.8% to 69.2% of the total particles and the highest PBAP fraction (69.2%) and HNA
concentration is observed on 4/14 ($5.25\times10^4 \pm 5.89\times10^3$ m$^{-3}$). The total FBAP fraction (1 to 5µm) ranges
from 16% to 43%, but it reaches its maximum on 4/15. However, ABC fraction of the total WIBS particle
concentration ranges from 1.3% to 9.2% and it reaches its maximum on 4/14. Even when the magnitudes
of the PBAP and FBAP fractions differ on average by a factor of ~ 2 throughout the sampling period, both
techniques agree an enhancement in the total biological particles takes place between 4/14 to 4/16. Given
the uncertainty of the two methodologies, it is remarkable that there is such agreement between WIBS and
FCM results.

Among the four meteorological categories, humid and warm days characterize for showing the

highest HNA, A type, AB type and ABC type concentrations suggesting that A and AB types may also be
related to wet-ejected fungal spores in Metro Atlanta; this possibly explains why the ABC fraction of the
total FBAP in 4/7 is not as high as on 4/14 and 4/15 (Figure 3b), and differs with the behavior observed by
the HNA population on 4/7. The bioLNA population does not show a correlation to any specific FBAP type
and shows it highest concentrations during dry and cold days. In addition, bioLNA concentrations are
anticorrelated with type B concentrations (Figure S19, correlation coefficient, r = -0.59; $R^2$ =0.30) during
dry (both cold and warm) days, when bioLNA dominates the total PBAP concentration. Given that type B
particles have been previously correlated to abiotic particles (e.g. black carbon) in urban environments (Yue





et al., 2017), bioLNA and type B anticorrelation suggests that bioLNA particles may in fact represent a
heterogeneous bioaerosol population. That bioLNA is not correlated with any FBAP type gives rise to two
possibilities: (1) if bioLNA population is mainly composed of bacteria or agglomerated bacteria, then it is
possible that they are detected by multiple FBAP types and is not attributed specifically to one of them; (2)
the intrinsic fluorescence of bioLNA particles is too low and a high fraction of them is abiotic. It is
challenging to determine what PBAP types each WIBS FBAP type is mainly detecting. Based on WIBS-
4A results in Metro Atlanta, ABC type detects wet-ejected fungal spores, but still unclear what PBAP types
are detect by the other FBAP types or if they just capture a high fraction of non-biological particles. FBAP
types and WIBS total particles correlations in Figure S17 show all FBAP types are correlated to WIBS total
particles, but ABC and AB types show the lowest correlations (type AB: $R^2$ =0.101; type ABC: $R^2$ =
0.1266).

Figure 8 shows FCM total PBAP (black line), ABC type (light green), FL1(Channel A; dark green
line) and total FBAP (blue line) concentrations, where the FL1 concentration ( [FL1] ) constitutes the sum
of the number concentrations of types A, AB, AC, and ABC ([FL1] = [A] + [AB] + [AC] + [ABC]; Gabey
et al., 2011; Healy et al., 2014). Throughout the April-May 2015 sampling events, total PBAP
concentrations (1 to 5µm) were mainly constrained between the FL1 and ABC type concentrations
suggesting FL1 and ABC type represent the upper and lower bound PBAP concentrations in Metro Atlanta,
respectively. It also important to highlight that FCM PBAP concentrations are closer to the ABC type
concentrations before April 16 when the HNA population dominates, but then after April 16 FCM PBAP
concentrations are closer to FL1 concentrations when bioLNA starts to dominate the total PBAP
concentration. In addition, Figure 8 shows that total FBAP (sum of type A, B, C, AB, AC, ABC) exceeds
the (corrected) PBAP concentrations in Metro Atlanta.

## 5. Conclusions

In this study we presented the development and testing of an effective FCM protocol to identify and
quantify bioaerosol populations. The FCM protocol, designed to constrain any particle accumulation due
to cleaning or by fluid supplies, successfully quantified the day-to-day variability of bioaerosols in the
Atlanta Metro area. It is the first FCM study to detect well-defined LNA (low nucleic acid) and HNA (high
nucleic acid) atmospheric biological populations under different meteorological scenarios. FCM results
show dynamic bioaerosol populations in Atlanta leading to a 84.0% of HNA (wet-ejected fungal spores)
and 6.1% bioLNA contribution to the PBAP number (1 to 5µm range), respectively, during humid and
warm days after rain events. However, bioLNA dominates warm and cold dry days, constituting 72% of
the PBAP number concentration.



WIBS-4A and SpinCon II collocated sampling showed that the HNA and ABC type concentrations are

well correlated ($R^2$=0.40) and display similar size distribution. We therefore conclude that both instruments
detect the same particles, and used empirical collection/detection efficiency factors to correct the FCM size
distributions and concentrations in the 1 to 5µm diameter range. WIBS-4A and FCM results suggest Metro
Atlanta PBAP concentrations range between $10^4$ - $10^5$ m$^{-3}$ (1 to 5µm) and they can constitute a substantial
fraction of coarse mode particle concentration (WIBS-4A: 43%; FCM: 69%), comparable to the PBAP
coarse mode fraction in highly vegetated environments. The FCM bioLNA population, possibly containing
bacterial cells, did not correlate to any FBAP type. The fact that the bioLNA population is not correlated
with a specific FBAP type suggests it may be particularly challenging to use LIF techniques to distinguish
bioaerosols with low intrinsic autofluorescence from non-biological particles, especially given the
heterogeneities introduced by the large biodiversity of airborne microbes. The possible influence of abiotic
particles in the bioLNA population can also explain the lack of correlation between bioLNA and FBAP
types given that the FCM threshold approach does not ensure total exclusion of abiotic particles. In addition,
the unspecific binding of SYTO-13 to abiotic particles cannot be ruled out in the bioLNA population. FCM
comparison between atmospheric and pure culture samples showed lower SYTO-13 fluorescence
intensities in the atmospheric samples and suggests a degradation in the genetic material of PBAP, possibly
caused by the limited nutrients and strong stress prevailing in the atmosphere, which further challenge the
ability of LIF to distinguish bioLNA.

In summary, this study have shown for the first time that FCM can effectively identify, quantify and

study the daily variability of heterogeneous PBAP populations (e.g. HNA, bioLNA and pollen) with
different genetic material content in an urban environment to the degree of quantitatively correlate FCM
HNA to WIBS-4A ABC type number concentrations and better understand wet-ejected fungal spores
enhancement after rain events. Furthermore, FCM and WIBS-4A results show bacterial cells detection and
quantification still a challenging task for LIF technology as well as for FCM given the complexity involved
to minimize abiotic interferences, and to the heterogenicity of the atmospheric samples.

**Acknowledgments**
We acknowledge support from a Georgia Power Faculty Scholar chair, a Cullen-Peck Faculty Fellowship,
a Dreyfus Foundation Postdoctoral Fellowship in Environmental Chemistry, NASA, and a NASA Earth
System Science Fellowship. We also thank Prof. Rodney Weber for helpful suggestions on the SpinCon II
flow calibration.



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





**Table 1:** Summary of the SpinCon II sampling events, the 24 h. averaged RH, ambient temperature, the
assigned meteorological category (using Section 4.4 definitions) and the corrected FCM-derived PBAP
number concentration (1 to 5 μm) for each sample collected during this study.

| Date (starting – ending time) | RH (%) | Temperature (°C) | Meteorological Category | PBAP Concentration (m$^{-3}$) 1 to 5μm diameter range |
|---|---|---|---|---|
| 4/7/15   (11:17 - 15:17) * | 70.9 | 21.4 | Humid, Warm | $9.282 \times 10^4$ |
| 4/8/15   (11:10 - 15:10) | 53.6 | 24.9 | Dry, Warm | $5.203 \times 10^5$ |
| 4/9/15   (11:15 - 15:15) | 53.8 | 25.3 | Dry, Warm | $1.254 \times 10^5$ |
| 4/14/15 (11:30 - 15:30) * | 76.8 | 22.5 | Humid, Warm | $8.253 \times 10^4$ |
| 4/15/15 (11:40 - 15:40) * | 83.6 | 18.9 | Humid, Warm | $1.234 \times 10^5$ |
| 4/16/15 (10:55 - 14:55) | 86.3 | 12.5 | Humid, Cold | $3.399 \times 10^5$ |
| 4/21/15 (13:15 - 17:15) | 43.2 | 16.6 | Dry, Cold | $4.741 \times 10^5$ |
| 4/22/15 (11:25 - 15:25) | 41.2 | 19.0 | Dry, Warm | $3.351 \times 10^5$ |
| 4/23/15 (11:35 - 15:35) | 48.1 | 16.8 | Dry, Cold | $1.708 \times 10^6$ |
| 4/28/15 (12:25 - 16:25) | 45.3 | 17.0 | Dry, Cold | $4.899 \times 10^5$ |
| 4/29/15 (11:55 - 15:55) # | 79.4 | 14.2 | Humid, Cold | $4.591 \times 10^5$ |
| 4/30/15 (12:10 - 16:10) | 57.3 | 17.4 | Dry, Cold | $9.603 \times 10^5$ |
| 5/13/15 (10:50 - 14:50) | 40.1 | 23.5 | Dry, Warm | $3.680 \times 10^5$ |
| 5/14/15 (11:50 - 15:50) | 52.3 | 23.0 | Dry, Warm | $4.851 \times 10^5$ |
| 5/15/15 (10:19 - 14:19) | 64.4 | 23.1 | Dry, Warm | $1.656 \times 10^6$ |

* Sampling occurred post-rain event.
# Sampling occurred during a rain event.



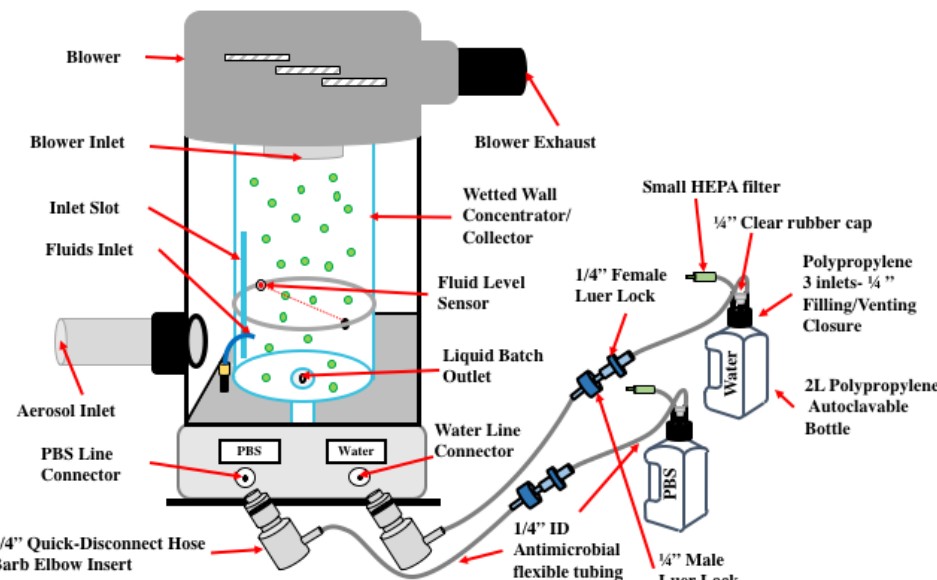

**Figure 1**: SpinCon II sampling setup including modified fluid supply system with anti-microbial tubing
and 2L Autoclavable bottles.





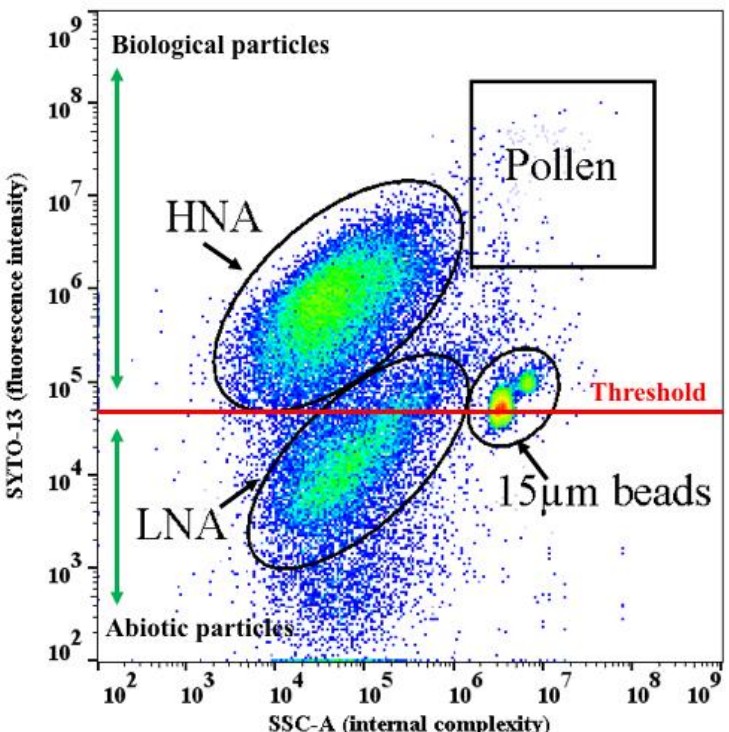

**Figure 2:** FL1-A vs. SSC-A plot used to identify populations in collected rooftop atmospheric samples
(April 14, 2015 4hr sample) including: the 42k threshold line in red and, abiotic particles (below
threshold) and biological particles (above threshold) designated regions. The fraction of the LNA
population above the threshold line is referred as the "bioLNA" population.





**Figure 3:** WIBS-4A 4h averaged results of WIBS total particle, NON-FBAP, total FBAP and type ABC and FBAP fraction in the right Y-axis in (a) and FBAP types number concentration fractional composition in (b); and average 1 to5µm size distributions (average of the 15 sampling events 4h average) over the 15 SpinCon II sampling events of WIBS total particles and NON-FBAP in (c) and all FBAP types, except AC type in (d). AC type showed low statistics and constituted less than 1% of the total FBAP (not shown)






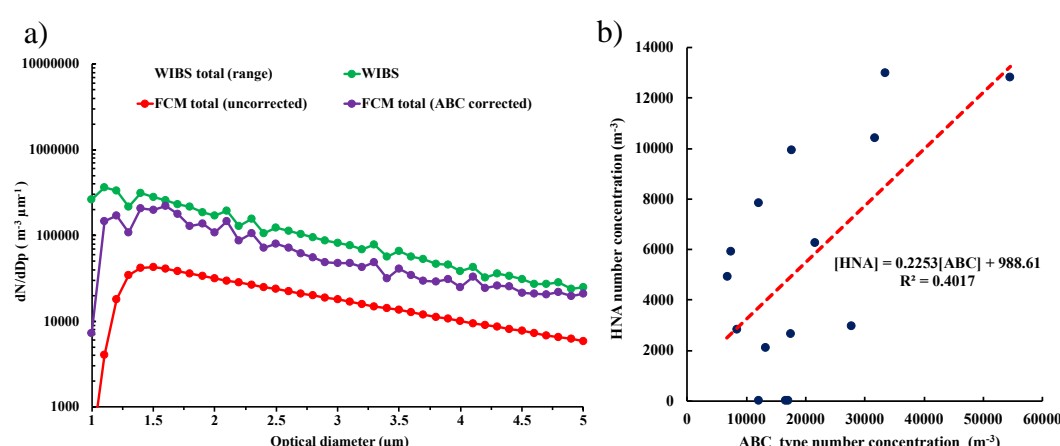

**Figure 4:** WIBS-4A, FCM uncorrected and FCM (ABC corrected) total particle concentration (1 to 5µm)
average size distributions (geometrically averaged over the 15 SpinCon II sampling events) including WIBS
range (± geometric standard deviation factor) in (a); and HNA and ABC type concentration correlation in
the 1 to 5µm range in (b) including it linear correlation in red.



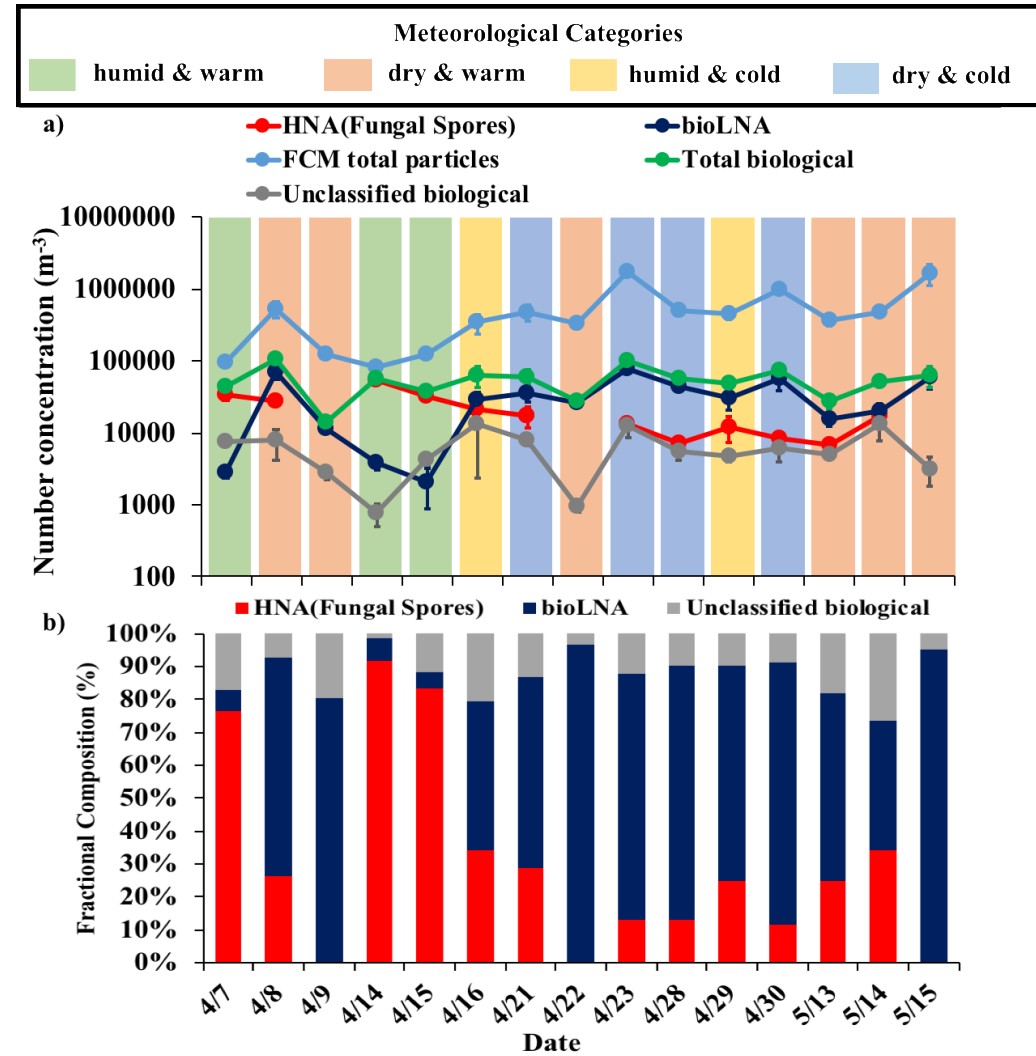

**Figure 5:** FCM total particle, HNA, bioLNA and total PBAP number concentrations in the 1 to 5µm
range highlighting the prevailing meteorological category during each sampling event in (a); HNA and
bioLNA number concentration fractional compositions for each sampling event in (b).






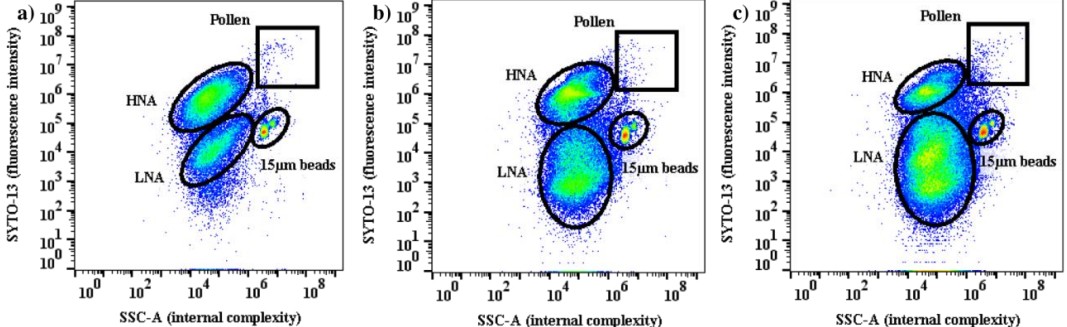

**Figure 6:** FL1-A vs. SSC-A FSC plots for (a) April 14, (b) April 15, and, (c) April 16. This period was
characterized by a transition from humid & warm to humid & cold conditions (diurnal average RH=77%,
T=22.5 °C on 4/14; RH=84%, T=18.9 °C on 4/15, and RH= 86%, T= 12.5 °C on 4/16). The FCM plots
during this transition period show a decrease of fungal population and an increase of the LNA population.
In each population, warmer colors represent higher particle concentrations.




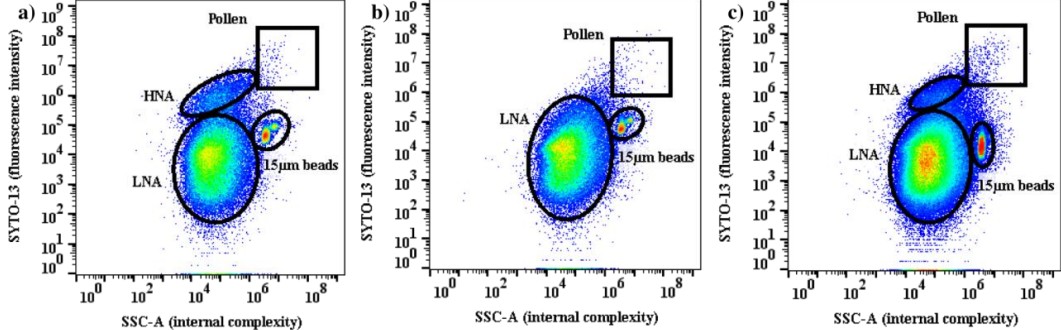

**Figure 7:** Similar to Figure 6, but for (a) April 21, (b) April 22, (c) April 23, which was characterized by
dry and variability in temperature (diurnal average RH=43%, T=16.6 °C on 4/21; RH=41%, T=19.0 °C on
4/22, and, RH= 48%, T= 16.8 °C on 4/23. Note the disappearance of the fungal spore population on the
warmest day (4/22).





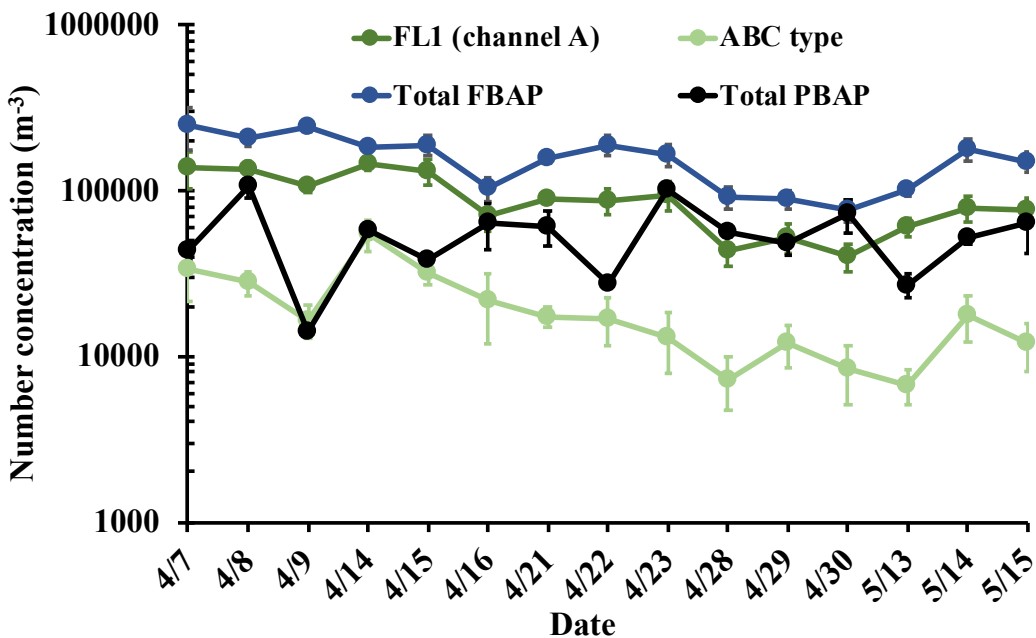

**Figure 8:** WIBS-4A total FBAP, FL1 and ABC type, and FCM total particle number concentrations in
the 1 to 5µm range for each sampling event from April 7 to May 15, 2015.
