# Peer review of "Using flow cytometry and light-induced fluorescence technique to characterize the"

_Atmospheric Chemistry and Physics, 2018_

## Referee Comment (RC1) · Anonymous Referee #1 · 21 Jan 2019

CONTEXT, ORIGINALITY and GENERAL COMMENTS This manuscript is attempting to answer to a need for the real-time resolution of whole cell microbial bioaerosols, using optical particle recognition (OPR) techniques. The authors present parallel application/assessment of aerosol cytometry (WIBS), conventional liquid cytometry (FCM) and direct microscopy (EPM). The scientific concepts supporting this manuscript—resolving the dynamic distributions of airborne microorganisms in "time and space"—is a reemerging research topic that has important environmental and public health implications for both indoor and outdoor environments. In this reviewer's opinion, this work

has potential to be a significant contribution to the bioaerosol literature at large; however, it needs significant technical revisions prior to publication in any forum, and in its essence is likely not a good fit for the mission of ACP. This disqualification is primarily attributed to the fact that this report is primarily an adaptive technology demonstration and comparison of existing analytical methods. Further, serial (unsupported) assumptions about microbial physiology are embedded in this manuscript, particularly with respect to identifying and quantifying the collected airborne microbiological agents based on presumed genomic characteristics. Important genomic characteristics have either been overly simplified or unfortunately omitted in critical contexts that are needed to support the heart of the work. These (over)simplifications and omissions make it difficult to sustain the author's conclusions given the data they acquired, presented and analyzed (juxtaposition of WIBS, FCM and EPM). The generalization that whole cell bioaerosols can be reliably deconstructed into two pools based on any non-normalized index of DNA/RNA content cannot be not supported by basic microbial (and plant) physiology and the data presented here.

The authors report that a "broad RH range" was monitored during these studies, and that their optical particle recognition data support the conclusion that markedly different microbial populations were airborne under the gross meteorological conditions defined here (T, RH). This may have been the case; however, the data reported are not compelling toward that end. The premise itself is tentative given the somewhat sensational statement that airborne microbes in a "broad" RH range were in-fact monitored, where 40's% < RH < 80's%. A majority of the observations reported (table 1) were under conditions near 50% RH ($\pm$ 9%); this RH is not near saturation conditions, nor is it near desiccating conditions; indeed, many would consider this a "midrange" of relative humidity. In this analytical context, (aut)ecological context or comparative environmental context, by no means is a couple of months of (bio)aerosol sampling conditions in Atlanta "ensuring a wide range of PBAP population(s), state(s) or concentration(s)" (page 6). The retinue of (new) aerobiological acronyms introduced by this manuscript make it difficult to follow at times; many of these (new) terms are not consistent with lexi-

con established over the last generation of bioaerosol research (or molecular biology research for that matter). This includes but is not limited to the following terms: LNA bioLNA HNA First and foremost, all intact (micro)biological cells contain nucleic acids, and the "bio" subscript prefix is conflicted with the fact that environmental nucleic acids can only be of biological origins, regardless of the "quantity" of nucleic acids inside any give (airborne) microbe. In this context, the authors did not acknowledge the fact that DNA is sequestered differently in bacteria, fungi, their spores and pollen grains; that this sequestration is sensitive to RH; and, that the configuration of intracellular DNA has tremendous implications for optical recognition methods and quantitation by FCM, regardless of genetic staining. Further, the authors did not acknowledge the fact that the range of DNA nucleotide pairs (106 – 107 per haploid genome) observed in bacteria, fungi and pollens may not be so different, that they can simply be relegated to "low" and "high" DNA pools by optical staining methods (see Alberts et al, Molecular Biology of the Cell, Section 8.6, Lack of relationship between amount of DNA and organism complexity). In this context, the identity and subsequent quantitation of microbial populations, like the optical DNA reporting methods used here, can be convoluted by the fact that microbial genomes, and common phylogenetic target sequences, are often sequestered intracellularly in markedly different conformations, gross lengths as well as in multiple copies. This is particularly variable across fungal genera where genomes (and their copies) can vary across two orders of magnitude (see Mohanta and Bae, (2015) The diversity of fungal genome. Biol Proced 17:8; and, Lofgren et al (2018), Genome-based estimates of fungal rDNA copy number variation across phylogenetic scales and ecological lifestyles, Molecular Ecology, 10.1111/mec.14995). To support their "low/high" DNA (genome) assignments, and associated microbial classifications, the authors should have, at a minimum, executed some (simple and inexpensive) DNA extractions on at least a subset of their aerosol samples, characterized sentinel sequences (basic qPCR) and juxtaposed this to their optical/cytometry data. In addition to the length/copy number variability presented above, in this midrange of relative humidity , DNA in airborne vegetative cells is in a transitional conformation, while that in

spores is held in a constant conformation regardless of RH (See Peccia et al, 2002, The Effect of Relative Humidity on the UV-induced Induced Inactivation of Airborne Bacteria, Aerosol Science and Technology 35(3):728). For these reasons, this reviewer suggests the work in this manuscript is not appropriate for the ACP readership, because in its essence, it is a comparative demonstration study of existing methods and instrumentation. Further, the manuscript suggests a correlation between fluorescence assignments and microbial phenotype, which are qualitative. For this and other reasons, this reviewer does not believe this is a good fit for the publication mission of ACP. This is does not mean, the work could not be reframed for publication in another journal which is more suited for this type of specialty work I would recommend the following options for the authors to publish this work: (i) condense the work into observation based manuscript (after major revision) as a demonstration study reduced to the constructive criticisms presented. I believe the manuscript has some potential for a good contribution to the bioaerosol literature, but should not be included in or near its present form in ACP (consider resubmission to: Aerosol Science and Technology or Journal of Aerosol Science or Atmospheric Environment).

Hernandez et al. (2016) is cited in text, however it is not presented in the bibliography, Perring et al. (2015) is cited in text, but not presented in bibliography

Specific Technical and Methodological Issues

While Spincon and its newer liquid impingement variants (SpinCon II and OMNI 30000) are recognized as a high volume sampler, these devices, like all impingers, impart a significant pressure drop and associated sampling stresses that are realized differently by bacteria, fungi their spores and pollen grains. While the specific characterization work of SpinConII by Kesavan and coworkers is appreciated, this does not mean the authors can simply dismiss collection stress and sampling efficiency differences, where it cannot be dismissed (Page 7) and the qualification of on (page 22) is convoluted for a reviewer skilled in this art; indeed the SpinCon II correction factors presented on page 22 are at odds with the statement on page 7. At a minimum he authors need

to acknowledge that fungal spores, because of their relative hydrophobicity, have a broad range of impingement retention potential, especially with respect to cells that are not in a sporulated forms (bacteria or fungi), and there are no correction factors that can specifically account for this unless the microbial community structure can be quantitatively apportioned with this equipment (see directly relevant now classic series on variability of bacteria and fungal spore collection/retention efficiency from University of Cincinnati Environmental Health group, circa 1996-2000):

Lin et al, (1999) Collection efficiency and culturability of impingement into a liquid for bioaerosols of fungal spores and yeast cells, Aerosol Science and Technology

Grinshpun et al, (1997) Effect of impaction, bounce and reaerosolization on the collection efficiency of impingers, Aerosol Science and Technology

Lin et al, (1999), Long-term sampling of airborne bacteria and fungi into a non-evaporating liquid, Aerosol Science and Technology

Lin et al, (1998) The effect of sampling time and flow rates on the bioefficiency of three fungal spore sampling methods, Aerosol Science and Technology

While the impingement, flow cytometry methods and DNA intercalating agents used are widely accepted, the their simple extension from aquatic environments (pages 3 and 7)) to generically understanding the "stresses" airborne microbes experience in aerosol environments does not directly support the authors analytical arguments or conclusions. This reviewer found the liquid collection, FCM analyses and microscopy juxtaposition presented by the authors convoluted as it relates to pollen analysis and the generic "two pool" DNA convention presented (i.g. bioLNA and LNA arguments where pollen assignments are concerned). The introduction of Ragweed spp. (page 16), the discussion of its physiology and fragmentation potential, only serves to complicate this matter and since the authors have no independent and definitive observations of pollen community structure from their sample sets, this review recommends condensing the text to omit any specific discussion of individual species. The same

criticism applies to arguments introducing and qualifying pure culture observations of yeasts (Saccharomyces cerevisiae) and bacteria (Pseudomonas aeruginosa) as germane to this investigation. Lastly, this reviewer finds the qualitative descriptions of fluorescence correlations to different microbial phenotypes, and any suggestion that WIBS can "speciate" different airborne microbes, unsupported and inappropriate (page 21). Clearly, the authors are skilled in descriptive statistics and have executed an exhaustive literature on fluorescence-based optical particle recognition instruments. To suggest ABC and HNA are "highly correlated" based on an $R2 = 0.4$ (figure 4), and that the AB type is "weakly correlated" with HNA where $R2 = 0.17$, is a subjective presentation of what should be objective thresholds (otherwise these authors need to present interpretive precedents in context and supported their qualitative arguments on why this is "highly" or "weakly" correlated).

―――――――――――――――――――――――――――

---

## Referee Comment (RC2) · Anonymous Referee #2 · 8 Mar 2019

This manuscript describes a protocol to investigate bioaerosols and compare different techniques. The authors use flow cytometry (FCM) protocol to identify different population of DNA-containing particles such as Low Nucleic Acid- content particles (bioLNA), High Nucleic Acid-content particles (HNA) within atmospheric bioparticles. They applied the protocol to study diversity and population of bioparticle in the Atlanta metro during various meteorological conditions. Further they access performance of Light Induced Fluorescence (LIF) for Fluorescent Biological Atmospheric Particles (FBAP) detection with FCM and Epifluorescence microscopy (EPM) techniques. The authors also

used a Wideband Integrated Bioaerosol Sensor (WIBS) and compared with FCM. They did not find any correlation between FBAP and bioLNA. In general, they suggested that it is challenging to detect bacterial cells. They found that HNA size distribution dominated by 3-5 micron particles and observed mostly high humid condition (RH>70%), suggested that HNA particles most likely correspond to fungal spores, probably wet discharged spores. While LNA size distribution ranges between 2 and 4 micron. The authors suggested that bacteria may contribute to the LNA particles.

Overall, the manuscript is quite detailed but some of the discussion of the results needs substantial improvement. The manuscript can be significantly improved by reducing some of the unnecessary detail of the techniques/comparison and irrelevant introduction part and by discussing the findings in coherent way. The authors should focus more on the science part. For example, the authors discussed correlation between techniques but in my opinion, the overall story is missing. Some of the claims need more support or better discussion. For example, detection of bacterial cells and pollen fragmentation using different methods are not convincing. Also the authors investigated diversity of bioparticles at different meteorological condition. They could look at the histograms of the relative humidity and temperature and see if there is any relation with the fungal spores or pollen fragments. Then the size distributions will help to understand at those conditions and relate to different bioparticles. Some suggested clarifications are listed below.

Line 510: LNA size distributions are dominated by 2-4 $\mu$m particle. Authors suggested that bacteria can contribute to this group. Are you sure about that? I believe bacteria are smaller in size. Line 497: Authors discussed about pollen cluster of the FCM results in Figure 2. It is not clear to me the pollen cluster. I don't see a clear cluster. Line 527: The authors suggested that pollen fragmentation will have negligible effect on LNA concentrations. However, previous studies suggested that pollen grain can rupture into many fragments. I am not sure about Ragweed pollen but different species of pollen rupture at high humid condition. If FCM protocol is used as a tool for detection

and quantification of bioparticle in other location where different species of pollen are present. Then how should we interpret the FCM data? Line 532: How did you compare the pollen concentrations and LNA concentrations? Line 539: How do you get the size information in Figure 2? Discussion of figure 2 and 3 needs improvement. Line 560: is it possible that "unclassified" bioparticles contribute from secondary bioparticles such as fragments from fungal spores and pollen? Fragmented particles might have broad size distributions and may change their chemistry?

---

## Author Comment (AC1) · 3 Jun 2019

**Response to Reviewer 1**

*First of all, we want to thank the reviewer #1 for meticulously reading our manuscript and for providing the critical review to improve the manuscript. Below, we include the response to comments and concern of reviewer #1.*

**Reviewer comment:** "Further, serial (unsupported) assumptions about microbial physiology are embedded in this manuscript, particularly with respect to identifying and quantifying the collected airborne microbiological agents based on presumed genomic characteristics. Important genomic characteristics have either been overly simplified or unfortunately omitted in critical contexts that are needed to support the heart of the work. These (over)simplifications and omissions make it difficult to sustain the author's conclusions given the data they acquired, presented and analyzed (juxtaposition of WIBS, FCM and EPM). The generalization that whole cell bioaerosols can be reliably deconstructed into two pools based on any non-normalized index of DNA/RNA content cannot be not supported by basic microbial (and plant) physiology and the data presented here."

*Response: SYTO-13 stains DNA and RNA nucleic acids (Lebaron et al.,2001; Troussellier et al.,1999; Comas-Riu et al., 2002; https://www.thermofisher.com {Cell-Permeant Cyanine Dyes: The SYTO Nucleic Acid Stains}), and the resulting fluorescence intensity is directly related to the nucleic acid content. Previous literature clearly shows that SYTO-13 can effectively distinguish between HNA and LNA bacterioplankton and phytoplankton populations in fresh and seawater environments, and results are comparable to SYBR green II and SYBR green I, more specific DNA probes (Lebaron et al., 2001; Bouvier et al., 2007). Furthermore, the genome size of sorted HNA and LNA populations in fresh water have shown HNA populations will likely have a larger genome than LNA populations (Schattenhofer et al., 2011). That said, we do not claim that specific types of airborne microorganisms (e.g. bacteria, fungal spore, pollen) were quantified based on the staining intensity, as genome sizes of bacterial and fungal spores may overlap. However, the FCM also detects the physical particle size, which is considerably different between bacteria and fungal spores. Size and fluorescence intensity combined then allow the differentiation, which we denote as the low nucleic acid (LNA) and high nucleic acid (HNA) populations. Therefore, this distinction of stained bioparticles appears to be robust.*

*Atmospheric samples are different from aquatic samples in composition and particle sizes, but overall the classification of the HNA and LNA populations is based on the fact that SYTO-13 directly stains nucleic acids and it is well established and accepted in the flow cytometry community. Furthermore, quantifying the DNA and RNA content of specific HNA and LNA populations to determine, for instance, which are bacterial vs. fungal, constitutes the design and optimization of a protocol for in-situ sorting, and quantification, and subsequent molecular analysis of the sorted populations, which is the material for another publication. This should be the next step to have a more specific FCM microbial quantification, but in no way should invalidate our conclusions.*

**Reviewer Comment:** "The premise itself is tentative given the somewhat sensational statement that airborne microbes in a "broad" RH range were in-fact monitored, where 40's% < RH < 80's%. A majority of the observations reported (table 1) were under conditions near 50% RH ($\pm$ 9%); this

RH is not near saturation conditions, nor is it near desiccating conditions; indeed, many would consider this a "midrange" of relative humidity. In this analytical context, (aut)ecological context or comparative environmental context, by no means is a couple of months of (bio)aerosol sampling conditions in Atlanta "ensuring a wide range of PBAP population(s), state(s) or concentration(s)" (page6)."

*Response: Observations are indeed limited to 15 sampling events conducted in Atlanta, GA during Spring 2015, and we have edited our statements to more precisely reflect what was performed. 24 hour averaged temperature and relative humidity were calculated in order to determine the prevailing temperature and relative humidity (RH) during each sampling event given that meteorological conditions during the sampling time (4hr average) will not necessary represent the meteorological conditions of the whole sampling day. In addition, the residence time of microorganisms (e.g. bacteria and fungal spores) in the atmosphere is larger than SpinCon II sampling time (4hr), which means microorganisms aerosolized the night before or hours before sampling started could still be collected (Delort and Amato, 2018 – Microbiology of Aerosols-Section2.3.4: Residence time, transport history, and emission models). However, the temperature and relative humidity did vary during these 15 sampling events, as shown in the table below. Humid and warm days (April-7: Max RH- 97%, April-14: Max RH-93%, April-15: Max RH-91%) after rain events observed max RH above 90%. In contrast, multiple dry days (e.g. April-8, April-21, April-22, May-13) experienced minimum RH below 30%. It is important to highlight the main reason of the RH and temperature categories is to better understand the substantial change in composition observed between the LNA and HNA populations between sampling events, but we cannot rule out the rain events and soil wetness possible role in the enhancement of the HNA population on April-7, April-14 and April-15.*

*Table1*: 24 hr. relative humidity and temperature average(Avg.), minimum(Min), maximum(Max)

| Days | 24hr. Avg.Temperature (°C) | Min (°C) | Max (°C) | 24 hr. avg. Relative Humidity (%) | Min (%) | Max(%) |
|---|---|---|---|---|---|---|
| 7-Apr | 21.4 | 16.7 | 26.8 | 70.9 | 40.0 | 97.0 |
| 8-Apr | 24.9 | 17.9 | 31.2 | 53.6 | 26.0 | 84.0 |
| 9-Apr | 25.3 | 20.4 | 30.3 | 53.8 | 35.0 | 76.0 |
| 14-Apr | 22.5 | 19.1 | 28.7 | 76.8 | 47.0 | 93.0 |
| 15-Apr | 18.9 | 12.8 | 24.7 | 83.6 | 60.0 | 91.0 |
| 16-Apr | 12.5 | 11.3 | 13.7 | 86.3 | 80.0 | 94.0 |
| 21-Apr | 16.6 | 10.4 | 22.1 | 43.2 | 19.0 | 75.0 |
| 22-Apr | 18.8 | 11.6 | 26.1 | 38.1 | 19.0 | 60.0 |
| 23-Apr | 16.8 | 13.9 | 19.6 | 48.1 | 27.0 | 77.0 |
| 28-Apr | 17.0 | 12.8 | 21.8 | 45.3 | 34.0 | 66.0 |
| 29-Apr | 14.2 | 12.0 | 16.9 | 79.4 | 63.0 | 89.0 |
| 30-Apr | 17.4 | 11.3 | 23.7 | 57.3 | 28.0 | 90.0 |
| 13-May | 23.5 | 16.7 | 30.1 | 40.1 | 20.0 | 63.0 |
| 14-May | 23.0 | 18.3 | 28.0 | 52.3 | 39.0 | 63.0 |
| 15-May | 23.1 | 19.8 | 25.8 | 64.4 | 53.0 | 81.0 |

**Reviewer comment:** "This includes but is not limited to the following terms: LNA, bioLNA, HNA. First and foremost, all intact (micro)biological cells contain nucleic acids, and the "bio" subscript prefix is conflicted with the fact that environmental nucleic acids can only be of biological origins, regardless of the "quantity" of nucleic acids inside any give (airborne) microbe. In this context, the authors did not acknowledge the fact that DNA is sequestered differently in

bacteria, fungi, their spores and pollen grains; that this sequestration is sensitive to RH; and, that the configuration of intracellular DNA has tremendous implications for optical recognition methods and quantitation by FCM, regardless of genetic staining."

*Response: The "bioLNA" population highlights the fraction of particles in the LNA population above the autofluorescence threshold value (42k). As a result, we can denote the "bioLNA" population as "LNA-AT" (LNA above threshold) from now on. The DNA sequestration by bacteria, fungal spores and pollen may differ and their cell membrane characteristics will ultimately determine how much stress the cells will sustain before they completely rupture. SYTO-13 is a highly permeable stain and have shown to be effective to detect nucleic acids (DNA and RNA) of bacteria endospores and vegetative cells (Comas Riu et al.,2002). Also, all pure cultures studied during this study are effectively stained by SYTO-13. Fungal spores have also been effectively stained by DNA/RNA probes (Bochdansky et al., 2017; Chen and Li et al.,2005), but we acknowledge in the revised manuscript that some fungal spores might not be equally stained due to their harder cell wall, and chromatin-binding of DNA.*

*Pollen can fragment at high RH and possibly be part of the LNA population as we have suggested in the manuscript, but these fragments will likely be below 1µm (Bacsi et al., 2006). Recently Santl-Temkiv et al.2017 observed bacteria cultivability is maintained (80% cultivable based on CFU counts), but leucine uptake rate (to measure metabolic activity) is reduced after 1hr sampling in the Spincon suggesting cells will be in a dormant state after 4hr sampling in the SpinCon II. Airborne microbes may also be stressed upon collection so it is possible that the LNA and HNA populations are two distinctive populations given that no anticorrelation is observed between the geometric mean fluorescence of the two populations. Based on Bouvier et al.2007, cell populations with different metabolic activity (e.g. active and non-active), when detected by FCM, should observe a decrease in fluorescence intensity in consecutive sampling events if transition from the HNA to the LNA population, or vice-versa if transition from LNA to HNA population. The fluorescence intensity of the LNA and HNA populations show small variation throughout the sampling events (BioLNA: $7.38 \times 10^4 \pm 1.39 \times 10^4$; HNA: $6.72 \times 10^5 \pm 2.30 \times 10^5$ ) and no anticorrelation is observed in the studied parameters (FSC-A, SSC-A, FL1-A), which supports we have in fact two distinctive population of bioaerosols (look Figure 1 below; Also look Figure S15 in the supplemental information).*

[Figure]

*Figure 1*: FL1-A fluorescence intensity of the BioLNA and HNA populations during the 15 sampling events. No HNA population identified on 4/9, 4/22, 5/15. Standard deviation of the fluorescence intensity is negligible for both populations throughout all sampling events.

**Reviewer comment: "**To support their "low/high" DNA (genome) assignments, and associated microbial classifications, the authors should have, at a minimum, executed some (simple and inexpensive) DNA extractions on at least a subset of their aerosol samples, characterized sentinel sequences (basic qPCR) and juxtaposed this to their optical/cytometry data. In addition to the length/copy number variability presented above, in this midrange of relative humidity spores is held in a constant conformation regardless of RH…"

*Response: Certainly, DNA extraction and sequencing of the atmospheric samples would allow the identification of specific bacteria, fungi taxonomical groups in the samples and their respective relative abundances. They are less effective when compared against FCM results, as it is unclear how the DNA is sufficiently different between the HNA and LNA populations. Sorting and subsequent DNA extraction of the sorted populations could be the path to determine the composition of the HNA and LNA population, but we could have limited biomass content to perform DNA sequencing of each population. In addition, qPCR quantification would not be directly comparable to FCM concentrations because bacteria (1 to 15 gene copies per cell) and fungi (30 to 100 gene copies per cell) ribosomal RNA gene copies vary depending the species considered (The ribosomal RNA operon copy number database, https://rrndb.umms.med.umich.edu/about/ ; DeLeon-Rodriguez et al., 2013). Then, to perform quantification an average copy number per cell has to be assumed, which can affect qPCR quantification by up to two orders of magnitude. The main point is that the corroboration of the HNA and LNA population through DNA extraction and sequencing would need to include effective sorting of the populations to be conclusive and thus, should be the subject of a future study/manuscript.*

**Reviewer comment:** "While the specific characterization work of SpinCon II by Kesavan and coworkers is appreciated, this does not mean the authors can simply dismiss collection stress and sampling efficiency differences, where it cannot be dismissed (Page 7) and the qualification of on (page 22) is convoluted for a reviewer skilled in this art; indeed the SpinCon II correction factors presented on page 22 are at odds with the statement on page 7."

*Response: We understand the point raised – and have also been considering the effects of long sampling times on the integrity of the cell membrane. The correction factor derived by comparing the WIBS and FCM size distributions is consistent with the Kesavan et al.2015 results, whom conducted shorter time sampling (< 30 min) than ours (4 hours). The estimated overall sampling efficiency is lower than Kesavan et al.2015, which means additional particle losses mechanisms are important during long sampling events (look Figure S12b in the supplemental information).*

**Reviewer comment:** "While the impingement, flow cytometry methods and DNA intercalating agents used are widely accepted, their simple extension from aquatic environments (pages 3 and 7)) to generically understanding the "stresses" airborne microbes experience in aerosol environments does not directly support the authors analytical arguments or conclusions"

*Response: Microbial cells in both environments could be under starvation given the limited amount of nutrients compared to pure culture liquid media. Furthermore, given SYTO-13 fluorescence intensity is directly related to the amount of nucleic acids in cells we performed a direct comparison between the atmospheric sample populations and pure culture populations, but we understand the LNA and HNA may represent a mixture of different types of cells and by no mean we aim to identify a specific microbial population in the atmospheric samples through this comparison. The main goal of the pure culture experiments in this manuscript is to serve as positive controls to ensure SYTO-13 effectively stains bacteria, fungi and pollen, and have reference fluorescence and scattering properties of each population.*

**Reviewer comments:** "This reviewer finds the qualitative descriptions of fluorescence correlations to different microbial phenotypes, and any suggestion that WIBS can "speciate" different airborne microbes, unsupported and inappropriate (page 21). Clearly, the authors are skilled in descriptive statistics and have executed an exhaustive literature on fluorescence-based optical particle recognition instruments. To suggest ABC and HNA are "highly correlated" based on an R2 = 0.4 (figure 4), and that the AB type is "weakly correlated" with HNA where R2 = 0.17, is a subjective presentation of what should be objective thresholds (otherwise these authors need to present interpretive precedents in context and supported their qualitative arguments on why this is "highly" or "weakly" correlated)."

*Response: First, we do not claim the WIBS-4 can speciate between different airborne microbes; we do however observe similar behavior between the FCM HNA population and ABC type particles, especially during humid and warm days after rain events (4/7, 4/14, 4/15). Also, we observed a moderately strong correlation ($R^2 = 0.40$; p-value = 0.016) between HNA and ABC type concentrations as well as similar size distributions between both populations. Compared to previous literature our level of correlation is comparable to those observed by Healy et al.2014 between microscopy quantification and WIBS-4 measurements. We also understand Gosselin et*

*al.2016 observed stronger correlations between fungal spores (inferred from mannitol and arabitol concentrations) and WIBS-4 concentrations, but that may just be because our studies were carried out in completely different environments (Rocky mountains vs. polluted urban environment). Our results therefore suggest WIBS-4 ABC type and FCM HNA population correspond to wet-ejected fungal spores on humid and warm days after rain events. As additional supporting information, the figure below shows the enhancement in the AB and ABC type concentration right after the beginning of the rain event on 4-13-15 (6pm; not correlated to NON-FBAP concentration), FBAP concentration enhancement previously linked to wet-ejected fungal spores (Huffman et al., 2013; Gosselin et al., 2016). Similar FBAP enhancement is observed during the rain events before sampling on 4/7 and 4/15.*

[Figure]

*Figure 2:* WIBS AB and ABC type concentration enhancement during rain events between 4/13 to 4/14. Includes high resolution temperature(yellow), relative humidity(blue) and rain rate(purple) measurements taken in the ES&T rooftop.

**Additional References:**

Bochdansky, A. B., Clouse, M. A., and Herndl, G. J.: Eukaryotic microbes, principally fungi and labyrinthulomycetes, dominate biomass on bathypelagic marine snow, The Isme Journal, 11, 362, 10.1038/ismej.2016.113, 2016.

Comas-Riu, J. and Vives-Rego, J. (2002), Cytometric monitoring of growth, sporogenesis and spore cell sorting in *Paenibacillus polymyxa* (formerly *Bacillus polymyxa*). Journal of Applied Microbiology, 92: 475-481. doi:10.1046/j.1365-2672.2002.01549.x

Crawford, I., Ruske, S., Topping, D. O., and Gallagher, M. W.: Evaluation of hierarchical agglomerative cluster analysis methods for discrimination of primary biological aerosol, Atmos. Meas. Tech., 8, 4979-4991, https://doi.org/10.5194/amt-8-4979-2015, 2015.

Hernandez, M., Perring, A. E., McCabe, K., Kok, G., Granger, G., and Baumgardner, D.: Chamber catalogues of optical and fluorescent signatures distinguish bioaerosol classes, Atmos. Meas. Tech., 9, 3283-3292, https://doi.org/10.5194/amt-9-3283-2016, 2016.

Perring, A. E., et al. (2015), Airborne observations of regional variation in fluorescent aerosol across the United States, *J. Geophys. Res. Atmos.*, 120, 1153–1170, doi:10.1002/2014JD022495.

Delort and Amato, (2018), Microbiology of aerosols, 1st Edition, pp.167-168.

Troussellier, M., Courties, C., Lebaron, P., and Servais, P.: Flow cytometric discrimination of bacterial populations in seawater based on SYTO 13 staining of nucleic acids, FEMS Microbiology Ecology, 29, 319-330, https://doi.org/10.1016/S0168-6496(99)00026-4, 1999.

Schattenhofer, M., Wulf, J., Kostadinov, I., Glöckner, F. O., Zubkov, M. V., and Fuchs, B. M.: Phylogenetic characterisation of picoplanktonic populations with high and low nucleic acid content in the North Atlantic Ocean, Systematic and Applied Microbiology, 34, 470-475, https://doi.org/10.1016/j.syapm.2011.01.008, 2011.

---

## Author Comment (AC2) · 3 Jun 2019

**Response to Reviewer 2**

We thank reviewer for meticulously reading our manuscript and for providing a thorough and thoughtful review. Our responses to the issues raised follow.

**Reviewer comment:** "Line 510: LNA size distributions are dominated by 2-4 µm particle. Authors suggested that bacteria can contribute to this group. Are you sure about that? I believe bacteria are smaller in size."

*Answer: Although many individual bacteria are likely in the order of ~ 1µm, the median aerodynamic diameter of culturable bacteria in continental sites has been ~ 4µm (Despres et al., 2012). Bacteria in the atmosphere can be co-emitted together with bigger particles (e.g. soil, plant fragments) and sometimes they are observed as clumps of bacteria cells (Burrows et al., 2009). For these reasons 2-4µm biological particles observed in the LNA population suggest large bacteria cells contribute to it. In addition, several bacterial species observed in the atmosphere (Microbiology of Aerosols, p.9; Monier and Lindow, 2003; Baillie and Read, 2001) are within this sizes range, like: Sphingomonas spp.(1.0-2.7µm), Methylobacterium spp. (1.0–8.0µm), Pseudomonas spp. (e.g. Pseudomona syringae, ~2.5µm) and Bacillus spp. (e.g. Bacillus anthracis, 3 - 10µm).*

*Therefore, we can not ensure the LNA population is solely composed by bacterial cells; the size range of the LNA population and epifluorescence microscopy results, however, support bacteria cells between 2 - 4µm contribute to the LNA population.*

*The above discussion will be reflected in the revised text.*

**Reviewer comment:** "Line 497: Authors discussed about pollen cluster of the FCM results in Figure 2. It is not clear to me the pollen cluster. I don't see a clear cluster."

*Answer: A pollen cluster is present in Figure 2, but it is not well defined because of small counting statistics (~200 counts) compared to the total counts (~50,000 counts). Pollen particles constitute less than 1% of the total particle number; given this, flow Jo cannot cluster it using the 2% contour plots (look Figure S3a). However, the pollen population showed very high autofluorescence when no SYTO-13 was added (look Figure S11 in the supplemental information), consistent with the literature (Pöhlker et al., 2012); given their autoflurescence, size and the low counts strongly points to pollen.*

**Reviewer comment:** "Line 527: The authors suggested that pollen fragmentation will have negligible effect on LNA concentrations. However, previous studies suggested that pollen grain can rupture into many fragments. I am not sure about Ragweed pollen but different species of pollen rupture at high humid condition. If FCM protocol is used as a tool for detection and quantification of bioparticle in other location where different species of pollen are present. Then how should we interpret the FCM data?"

*Answer: Although 0.2μm – 5μm pollen fragments can be generated upon rupture, pollen (e.g. Birch, Ryegrass, Oak, Olive) mainly breaks apart into submicron fragments by hydrolysis and favors fragmentation into small submicron (<1μm) particles (Taylor et al., 2002; Taylor et al., 2007; Bacsi et al., 2006; Grote et al., 2003) that are not considered in our FCM analysis. An additional factor to consider in pollen fragmentation is the number of fragments generated per pollen grain. Given pollen concentrations are 100-1000 times lower than bacteria concentrations in the atmosphere (Hoose et al.,2010), at least 100 supermicron (>1μm) pollen fragments will have to be released per pollen grain to considerably influence the LNA population, which has not been observed. This discussion, although mentioned in the supplementary material, will be further emphasized in the revised text.*

**Reviewer comment:** "Line 532: How did you compare the pollen concentrations and LNA concentrations?"

*Answer: Each pollen and LNA cluster defines their respective number concentration, and that was used for comparison; calculations were performed for each of the analytical triplicates. Comparisons was conducted without taking in consideration the threshold approach because it takes into account the whole LNA population, not just the "bioLNA"(LNA above threshold). On average pollen number concentration is 0.54 ± 0.48% of total LNA number concentration (min: 0.16%; max: 1.70%). After threshold application, pollen number concentration constitutes on average 1.70±1.36% (min: 0.36% max:4.06%) of the bioLNA number concentration. Overall, bioLNA number concentration (~$10^4$ $m^{-3}$) is two order of magnitude higher than pollen number concentration (~$10^2$ $m^{-3}$) throughout the 15 sampling events, and will be further discussed in the revised manuscript.*

**Reviewer comment:** "Line 539: How do you get the size information in Figure 2? Discussion of figure 2 and 3 needs improvement.

*Answer: This is a good point. In Figure 2 the FL1-A vs SSC-A plot shows the SYTO-13 fluorescence intensity vs. 90° scattering intensity (SSC-A; related to "internal complexity") for each single particle in a density plot; green and red zones denote the most populated regions. FSC-A value is related to particle size, and is determined based on a calibration (supplemental information, Figure S9, Equation S3) using standardized (e.g. 1μm, 2μm, 4μm,6μm,10μm &15μm) beads. Figure 2 does not show FSC-A-derived sizes, but we nevertheless report them.*

**Reviewer comment:** "Line 560: is it possible that "unclassified" bioparticles contribute from secondary bioparticles such as fragments from fungal spores and pollen? Fragmented particles might have broad size distributions and may change their chemistry?"

*Answer: The "unclassified" bioparticles are those not constrained by Flow Jo 2% contour gating, and most of these particles are far from the centroids of the gated populations. They can indeed be formed by fragmentation or accretion, or also be related to plant debris (i.e., irregular*

*bioparticles) that are characterized by a very broad size, internal complexity and nucleic acid content distributions. We will include these points in the revised manuscript.*

**Additional References:**

Baillie, L., and Read, T. D.: Bacillus anthracis, a bug with attitude!, Current Opinion in Microbiology, 4, 78-81, https://doi.org/10.1016/S1369-5274(00)00168-5, 2001.

Burrows, S. M., Elbert, W., Lawrence, M. G., and Pöschl, U.: Bacteria in the global atmosphere – Part 1: Review and synthesis of literature data for different ecosystems, Atmos. Chem. Phys., 9, 9263-9280, https://doi.org/10.5194/acp-9-9263-2009, 2009.

Hader, J. D., Wright, T. P., and Petters, M. D.: Contribution of pollen to atmospheric ice nuclei concentrations, Atmos. Chem. Phys., 14, 5433-5449, https://doi.org/10.5194/acp-14-5433-2014, 2014.

Grote, M., Valenta, R., and Reichelt, R.: Abortive pollen germination: A mechanism of allergen release in birch, alder, and hazel revealed by immunogold electron microscopy, J. Allergy Clin. Immun., 111, 1017–1023, doi:10.1067/mai.2003.1452, 2003.

Monier, J. M., and Lindow, S. E.: Pseudomonas syringae Responds to the Environment on Leaves by Cell Size Reduction, Phytopathology, 93, 1209-1216, 10.1094/PHYTO.2003.93.10.1209, 2003.

Taylor, P. E., Flagan, R. C., Valenta, R., and Glovsky, M. M.: Release of allergens as respirable aerosols: A link between grass pollen and asthma, Journal of Allergy and Clinical Immunology, 109, 51-56, https://doi.org/10.1067/mai.2002.120759, 2002.

**Revised Figure 2**

---

## Author Response (AR1)

[revised manuscript text omitted]
                      | RH   | Temperature | Meteorological | PBAP Concentration (m -3 ) |
|---------------------------|------|-------------|----------------|---------------------------------------|
| (starting – ending time)  | (%)  | (°C)        | Category       | 1 to 5µm diameter range               |
| 4/7/15 (11:17 - 15:17) *  | 70.9 | 21.4        | Humid, Warm    | 9.282×10 4                 |
| 4/8/15 (11:10 - 15:10)    | 53.6 | 24.9        | Dry, Warm      | 5.203×10 5                 |
| 4/9/15 (11:15 - 15:15)    | 53.8 | 25.3        | Dry, Warm      | 1.254×10 5                 |
| 4/14/15 (11:30 - 15:30) * | 76.8 | 22.5        | Humid, Warm    | 8.253×10 4                 |
| 4/15/15 (11:40 - 15:40) * | 83.6 | 18.9        | Humid, Warm    | 1.234×10 5                 |
| 4/16/15 (10:55 - 14:55)   | 86.3 | 12.5        | Humid, Cold    | 3.399×10 5                 |
| 4/21/15 (13:15 - 17:15)   | 43.2 | 16.6        | Dry, Cold      | 4.741×10 5                 |
| 4/22/15 (11:25 - 15:25)   | 41.2 | 19.0        | Dry, Warm      | 3.351×10 5                 |
| 4/23/15 (11:35 - 15:35)   | 48.1 | 16.8        | Dry, Cold      | $1.708 \times 10^{6}$                 |
| 4/28/15 (12:25 - 16:25)   | 45.3 | 17.0        | Dry, Cold      | 4.899×10 5                 |
| 4/29/15 (11:55 - 15:55) # | 79.4 | 14.2        | Humid, Cold    | 4.591×10 5                 |
| 4/30/15 (12:10 - 16:10)   | 57.3 | 17.4        | Dry, Cold      | 9.603×10 5                 |
| 5/13/15 (10:50 - 14:50)   | 40.1 | 23.5        | Dry, Warm      | 3.680×10 5                 |
| 5/14/15 (11:50 - 15:50)   | 52.3 | 23.0        | Dry, Warm      | 4.851×10 5                 |
| 5/15/15 (10:19 - 14:19)   | 64.4 | 23.1        | Dry, Warm      | 1.656×10 6                 |

\* Sampling occurred post-rain event. # Sampling occurred during a rain event.

---

## Referee Report (RR1)

Negron et al. submitted a manuscript titled "Using flow cytometry and light-induced fluorescence technique to characterize the variability and characteristics of bioaerosols in springtime at Metro Atlanta, Georgia." This manuscript presents a SpinCon/FCM protocol to identify and quantity bioaerosol populations and compares parallel data from an aerosol cytometry instrument (WIBS-4A). The research topic addresses emerging needs to improve the detection and identification of bioaerosols, which has an impact on several applications/communities. In general, I support the publication of this manuscript with some edits. There are suggestions for specific additions below, including some possibilities for added discussion and some suggestions.

Minor Comments:
- I suggest taking out the third paragraph entirely, lines 92-102. It doesn't seem to fit or add value in this section.
- Line 108- add the word "continuous"
  "[…] frequency measurements (~1 Hz) which make it ideal for *continuous* monitoring and […]"
- Lines 214-216: "[…] SpinCon has a better performance (product of the flow rate and the sampling efficiency) than any impingement sampler due to its high volumetric flow rate, which make it more suitable for bioaerosols detection (Kesavan et al., 2015)."
  The above statement is strong- cyclones are known to induce stress onto bioparticles and if identification and quantification is done by culture-based methods, then your collection process may result in low viability of the bioparticles collected. I suggest rephrasing this statement. I think the data comparison between the SpinCon/FCM and WIBS should be carefully reviewed.

Figure 3: I suggest using the same color scheme as Perring et al. 2015 for you WIBS information- this helps the WIBS community easily see the correlations between the particle types.

Major Comments
- Lines 520-522: Can you give more quantitative information on the differences of HNA concentrations on days 4/9, 4/22 and 5/15 compares to days with RH> 70%.
- Lines 639-654: As you mentioned in the introduction, fluorescence is size dependent- how is this factored into your analysis? You mentioned that Pollen > HNA> LNA-AT regarding fluorescence intensity, this is also true for the sizing of these particle assignments.
- Section 4.3: I think this section needs to consider the caveat of the collection approach of the Spin Con/FCM system vs the WIBS. As mentioned, the Spin Con is a cyclone collection approach, and therefore particles are subjected to a liquid, which can impact the fluorescence characteristics of a given particle depending on its chemistry. Whereas with the WIBS, the particles are not being 'collected', but rather just detected and is based on sheath flow. As a result, the fluorescence characteristics of the particle are not altered by 1) a harsh collection approach and 2) collection medium.
- Conclusion: are the authors suggesting that SpinCon/FCM provides better detection/identification than UV-LIF techniques? Given the caveat of the stress that the SpinCon induces on bioaerosols during the collection process- can this statement be made? Can the authors clearly state the advantages of the SpinCon/FCM over the current UV-LIF technology? What sparked the interest of the authors to use this introduced technique? Overall, I think this is an interesting study, however, I think the authors need to make it clear that this is a *complementary* analysis that the WIBS/UV-LIF may not provide. I do not think this is an

alternative approach to the detection/identification of bioaerosols, as I think there is more to explore with this technique.

Figure 2: From my understanding, Figure 2 displays fluorescence intensity versus particle shape information. What conclusions can the author draw from the information on this graph, for e.g. the intensity increases as the particle shape increases (not sure what an increasing SCC-A value means)- please explain more. In the WIBS-4A, there have been concerns/questions to how reliable the shape parameter. For e.g. the spatial alignment of the collection options (forward vs the 90 degree), the dynamic range of the detection, and even the angle at which a non-symmetrical particle hits the laser. The phrase "internal complexity" is a bit confusing when talking about the particle sphericity/shape- I suggest changing this phrase. Also, please explain in more detail what a higher value for SSC-A means- does it mean it is more spherical? Less spherical? And how does this help your suggestion on the populations/particle types you assigned in Figure 2? Overall, I suggest explaining more about the SSC-A parameter in FCM. Are you suggesting that pollen particles are more spherical than PSLs? Again, I think the SSC-A values need to be discussed in greater detail.

References:

1. Gabey, A. M., Gallagher, M. W., Whitehead, J., Dorsey, J. R., Kaye, P. H., and Stanley, W. R.: Measurements and comparison of primary biological aerosol above and below a tropical forest canopy using a dual channel fluorescence spectrometer, Atmos. Chem. Phys., 10, 4453–4466, https://doi.org/10.5194/acp10-4453-2010, 2010
2. Kaye, P. H., Aptowicz, K., Chang, R. K., Foot, V. E., and Videen, G.: Angularly resolved elastic scattering from airborne particles, Optics of Biological Particles, edited by: Hoekstra, A., Maltsev, V., and Videen, G., Springer, New York, USA, 31–61, 2007.

---

## Author Response (AR2)

**Response to Editor's comments:**

*We thank the editor for the comments that clearly improve the manuscript. In response, we have worked to condense the introduction of the manuscript to minimize unnecessary details, but still provide a thorough background for the readers. We have now included in the text most of the material provided in the responses to reviewer #1 and reviewer #2. Furthermore, taking into consideration the comments of reviewer # 3 and we have also given a more complete explanation in the manuscript of the SSC-A parameter used in Figure 2. Section 4.1 has also been modified to explain better figure 2 and to make it more readable. We kept the comparison of the pure culture experiments and the atmospheric populations, which can provide valuable information on the possible metabolic state of the cells and their relative FSC-A, SSC-A and FL1-A values. The supplemental information has also been expanded to include the information form the responses to reviewer #1. Also, Perring et al. (2015) color code have been adopted and Figure 3 has been modified accordingly to ensure readers understand WIBS results. Finally, all changes have been tracked in the manuscript and specify comments have been placed in the changes related to the responses to the reviewers.*

**Response to Reviewer #3 comments:**

*We want to thank the reviewer #for meticulously reading our manuscript and for providing the critical review to improve the manuscript. Below, we include the response to comments and questions raised.*

**Reviewer comment**: "I suggest taking out the third paragraph entirely, lines 92-102. It doesn't seem to fit or add value in this section".

*Answer: Lines 92-102 have been removed and the introduction has been revised to provide a more precise and readable research background. During the process some details in the introduction have been condensed as well.*

**Reviewer comment:** Lines 214-216: "[…] SpinCon has a better performance (product of the flow rate and the sampling efficiency) than any impingement sampler due to its high volumetric flow rate, which make it more suitable for bioaerosols detection (Kesavan et al., 2015)." The above statement is strong- cyclones are known to induce stress onto bioparticles and if identification and quantification is done by culture-based methods, then your collection process may result in low viability of the bioparticles collected. I suggest rephrasing this statement. I think the data comparison between the SpinCon/FCM and WIBS should be carefully reviewed.

*Answer: Thank you for raising this point. The statement has been rephrased accordingly in the revised manuscript (lines: 291-294).*

**Reviewer comment:** "Figure 3: I suggest using the same color scheme as Perring et al. 2015 for you WIBS information- this helps the WIBS community easily see the correlations between the particle types".

*Answer: The Perring et al. (2015) color scheme has been adopted and Figure 3 have been modified accordingly to ensure readers understand well WIBS results.*

**Reviewer comment:** "Lines 520-522: Can you give more quantitative information on the differences of HNA concentrations on days 4/9, 4/22 and 5/15 compares to days with RH> 70%".

*Answer: On 4/9, 4/22, 5/15 the HNA population is not identified. As a result, we consider their HNA concentration as 0 $m^{-3}$. During days with RH>70% (e.g. 4/7, 4/14, 4/15, 4/16, 4/29) concentrations range from $1.20 \times 10^4$ on 4/29 to $5.25 \times 10^4$ $m^{-3}$ on 4/14 (Figure 5a). Overall, the big difference in concentration when HNA is not identified may be related to the wet-ejected mechanism of specific fungal spores, emitted during specific meteorological (e.g. temperature, RH, rain events) and soil wetness conditions.*

**Reviewer comment:** "Lines 639-654: As you mentioned in the introduction, fluorescence is size dependent- how is this factored into your analysis? You mentioned that Pollen > HNA> LNA-AT regarding fluorescence intensity, this is also true for the sizing of these particle assignments".

*Answer: Autofluorescence may indeed increase as a function of size, but the fluorescence of stained particles may not – as seen, for example, for the HNA and LNA populations (Figure S10). Given the large heterogeneity and variability of the populations, we decided to adopt a conservative autofluorescence threshold approach that is not size dependent, but separates 99.5% of the PBAP particle number. We therefore treat each property independently. The application of a size dependent autofluorescence threshold will bring some challenges given that the autofluorescence of microorganisms may also depend on the composition and the metabolic state of the cells, which vary between each sampling event.*

**Reviewer comment:** "Section 4.3; consider caveats of the collection approach of the SpinCon/FCM system vs. the WIBS…"

*Answer: In Section 4.3 we consider several of the caveats of the comparison between SpinCon/FCM system and WIBS results, including that the comparison is restricted to 1 to 5µm size range. We also acknowledge SpinCon liquid sampling may stress cells (e.g. shrinking, expansion, bursting), affecting FCM size distribution. Changes has been made accordingly between lines 831 and 835 of the revised manuscript to acknowledge the caveats of the SpinCon liquid sampling.*

**Reviewer comment:** "Conclusion: are the authors suggesting that SpinCon/FCM provides better detection/identification than UV-LIF techniques? Given the caveat of the stress that the SpinCon induces on bioaerosols during the collection process- can this statement be made? Can the authors clearly state the advantages of the SpinCon/FCM over the current UV-LIF technology? What sparked the interest of the authors to use this introduced technique? Overall, I think this is an interesting study, however, I think the authors need to make it clear that this is a *complementary* analysis that the WIBS/UV-LIF may not provide. I do not think this is an alternative approach to the detection/identification of bioaerosols, as I think there is more to explore with this technique".

*Answer: The authors see FCM as a promising technology to perform a more specific detection of PBAP through direct staining of the nucleic acids (e.g. DNA/RNA) within the cells. Furthermore, understand future studies could sort populations to perform specific DNA sequencing of the identified/sorted populations as well as multiple probes could be included to the analysis of the sample to test the metabolic state (e.g. ATP production) and the viability of the cells. All the above constitute advantages of FCM over UV-LIF technology. We agree however that both methods combined give considerable amounts of information that each approach separately cannot and was one of the major conclusions of our manuscript. We have also emphasized this e.g., in Lines 1025 to 1029 in the revised manuscript.*

**Reviewer Comment:** "Figure 2… I suggest explaining more about the SSC-A parameter in FCM. Are you suggesting that pollen particles are more spherical than PSLs? Again, I think the SSC-A values need to be discussed in greater detail".

*Answer: The SSC-A scattering intensity is a function of particle size, the cellular granularity or density of the internal structures (e.g. nucleus, mitochondria, ribosomes), and sphericity of the particles (Mage et al., 2019; Mathaes et al., 2013). The size dependence of SSC-A is the reason why e.g., PSLs exhibit more scattering than pollen (Figure 2). Usage of side scattering intensity to obtain size is not common, however (Tzur et al., 2011). Side scattering has been effective to distinguish cells of different complexities (e.g. monocytes and granulocytes; Shapiro, 2005).*

*Sections 2.2 and 4.1 of the revised manuscript and the caption of Figure 2 is also modified to bring out the above points.*

**References:**

[revised manuscript text omitted]